

# A general theory on frequency and time-frequency analysis of irregularly sampled time series based on projection methods. II. Extension to time-frequency analysis

Guillaume Lenoir[1] and Michel Crucifix[1,2]

[1]Georges Lemaître Centre for Earth and Climate Research, Earth and Life Institute, Université catholique de Louvain, BE-1348, Louvain-la-Neuve, Belgium
[2]Belgian National Fund of Scientific Research, Rue d'Egmont, 5, BE-1000 Brussels, Belgium

*Correspondence to:* Guillaume Lenoir (guillaume.lenoir@hotmail.com)

**Abstract.** Geophysical time series are sometimes sampled irregularly along the time axis. The situation is particularly frequent in palaeoclimatology. Yet, there is so far no general framework for handling continuous wavelet transform when the time sampling is irregular.

Here we provide such a framework. To this end, we define the scalogram as the continuous-wavelet-transform-
equivalent of the extended Lomb-Scargle periodogram defined in part I of this study (Lenoir and Crucifix, 2017). The signal being analyzed is modeled as the sum of a locally periodic component in the time-frequency plane, a polynomial trend, and a background noise. The mother wavelet adopted here is the Morlet wavelet classically used in geophysical applications. The background noise model is a stationary Gaussian continuous autoregressive-moving-average (CARMA) process, which is more general than the traditional Gaussian white and red noise processes. The
scalogram is smoothed by averaging over neighboring times in order to reduce its variance. The Shannon-Nyquist exclusion zone is on the other hand defined as the area corrupted by local aliasing issues. The local amplitude in the time-frequency plane is then estimated with least-squares methods. We show that the squared amplitude and the scalogram are approximately proportional. Based on this property, we define a new analysis tool: the weighted smoothed scalogram, which we recommend for most analyses. The estimated signal amplitude also gives access to
band and ridge filtering. Finally, we design a test of significance for the weighted smoothed scalogram against the stationary Gaussian CARMA background noise, and provide algorithms for computing confidence levels, either analytically or with Monte Carlo Markov Chain methods. All the analysis tools presented in this article are available to the reader in the Python package WAVEPAL.

## 1   Introduction

The continuous wavelet transform (CWT) is widely used for the time-frequency analysis of geophysical time series, mainly through its scalogram, which is the squared modulus of the CWT. The CWT relies on an probing function, called the mother wavelet. A common choice for the mother wavelet is the Morlet wavelet (Grossmann and Morlet,





1984), which is well suited for the analysis of signals whose components have a time-varying frequency and/or an amplitude. The scalogram is then often smoothed to reduce its variance, and significance testing against a stationary Gaussian white or red noise is commonly applied. State of the art references in climate for the analysis of regularly sampled time series include: (Torrence and Compo, 1998) which provides the bases for the subsequent
works, (Torrence and Webster, 1999), providing a smoothing method for the scalogram (which is a particular case of the wavelet coherency developed in there), and (Maraun and Kurths, 2004; Maraun et al., 2007), which provide more reliable significance tests for the smoothed scalogram. A non-exhaustive list of the applications, in climatology, of the scalogram of the CWT with the Morlet wavelet, include:

– Studies in climate and weather: Analysis of the El Niño Southern Oscillation in (Torrence and Compo, 1998),
analysis of the Artic oscillation in (Grinsted et al., 2004), or the analysis of daily precipitations in the Alps in
        (Schaefli et al., 2007).

– Studies in paleoclimate: Analysis of the astronomical forcing in (Berger et al., 1998), analysis of the Mid-
        Pleistocene transition in (Elderfield et al., 2012), or the analysis of the equatorial Pacific thermocline over the
        last eight glacial periods in (Regoli et al., 2015).

Most of these studies use the algorithms provided by the papers cited above or similar algorithms, and all them require the data to be regularly spaced. However, it may happen that the time series be intrinsically irregularly sampled (this actually happens in some of the above examples) and it is then interpolated on a regularly spaced grid in order to apply the algorithms of the CWT and its scalogram. But the interpolation procedure may seriously affect the analysis with unpredictable consequences for the scientific interpretation, especially when performing significance
testing. This is illustrated in appendix F.

A solution to this problem was addressed by Foster in a series of articles which share a common thread with our two papers, in the sense that it first generalizes the Lomb-Scargle periodogram, based on orthognal projections methods, in (Foster, 1996a, b), and then extends the formalism to the continuous wavelet transform, in (Foster, 1996c), allowing not to interpolate the time series. Foster formulas were motivated by the astronomical study of the
light curves of variable stars, which are unevenly sampled time series with large gaps. The methods presented in this article are influenced by Foster theories and it is shown in appendix E that most of its formulas can actually be deduced from our general framework. The main limitations of Foster theory are the following (see appendix E for detailed explanations): significance testing is only performed for the white noise background case, it only deals with the unsmoothed scalogram, and the areas in the time-frequency plane corrupted by aliasing are underestimated. It
therefore suffers from a limited interest in geophysical applications[1]. Note that we have not found, in the literature, other rigorous methods tackling the problem of the estimation of the scalogram of irregularly sampled time series, based on an extension of the Lomb-Scargle periodogram, and without interpolating (explicitly or implicitly) the time series.

[1]An application of Foster formulas on paleoclimate data is found in (Witt and Schumann, 2005).





In this article, we extend the analysis tools that we derived in the first part of this study (Lenoir and Crucifix, 2017) in the case of the frequency analysis of irregularly sampled time series. They are based on a similar model, which is a locally periodic component in the time-frequency plane, plus a polynomial trend, plus a stationary Gaussian continuous autoregressive-moving-average (CARMA) process. Let us sketch the main points of the article. First, the

taper of the periodogram, derived in (Lenoir and Crucifix, 2017, Sect. 4.4), is chosen here to be a time-dependent Gaussian function with a variance depending on the scale, in order to define the Morlet wavelet-based scalogram. This is detailed in Sect. 3.2 of this work. Second, the scalogram is smoothed in order to reduce its variance, by averaging over neighboring times. To this end, we apply the same formula as in (Cohen and Walden, 2010). This is explained in Sect. 3.4. Third, in Sect. 3.5, we estimate the amplitude of the locally periodic component, extending

the results obtained in Sect. 6 of paper I, and define, in Sect. 3.6 of this article, the weighted smoothed scalogram as the time-frequency analogue of the weighted WOSA periodogram defined in the first part of this study (Lenoir and Crucifix, 2017). Fourth, we define in Sect. 3.8 the Shannon-Nyquist exclusion zone (SNEZ) to be the area of the time-frequency plane which must be excluded from the analysis because of the local aliasing issues. Fifth, we design a test of significance for the weighted smoothed scalogram, against the stationary Gaussian CARMA

background noise. This is based on the theory developed in Sect. 5 of paper I. More specifically, we define a null and alternative hypotheses, and estimate the distribution of the weighted smoothed scalogram under the null hypothesis, either analytically, conserving the first moments of the distribution, or with Markov Chain Monte Carlo (MCMC) methods. The latter approach allows to fully take into account the uncertainty on the parameters of the CARMA background process. This is presented in Sect. 4. Sixth, we provide, in Sect. 5, formulas for filtering the signal in

a band delimited by two scales, or with the ridges, which are the lines going through the maxima of the estimated amplitude, in the time-frequency plane. Ridge filtering is based on state of the art algorithms provided in (Lilly and Olhede, 2010) and (https://github.com/jonathanlilly/jLab). Seventh, we define in Sect. 6 the global scalogram as the time-averaged weighted smoothed scalogram, resulting in a periodogram-like analysis tool with a frequency-varying bandwidth. Eighth, we illustrate, in Sect. 7, the theory on the same paleoclimate data set as in our first article

(Lenoir and Crucifix, 2017). Finally, a Python package named WAVEPAL is available to the reader and is presented in Sect. 8. Before tackling the problem of irregularly sampled time series, the paper starts with the theory of the CWT applied to continuous-time signals. This gives the bases for the subsequent developments.

Most of the mathematical concepts and notations are introduced in the first part of this study (Lenoir and Crucifix, 2017), and the reader is invited to revise them. Throughout this article, we will denote the equations of the preceding

paper by e.g. "Eq. (I,30)", meaning "the equation (30) of paper I", and will refer to the paper itself by "paper I".





## 2  The continuous wavelet transform of continuous-time processes

### 2.1  The continuous wavelet transform and its scalogram

Mathematical background about Fourier analysis is given in appendix A. The continuous wavelet transform (CWT) of a function $x \in \mathbf{L}^2(\mathbb{R})$ is

$$S_x(\tau, a) = \langle \psi_{\tau,a} | x \rangle, \tag{1}$$

where $\psi_{\tau,a} \in \mathbf{L}^2(\mathbb{R})$ is defined by

$$\psi_{\tau,a}(t) = c(a) \psi\left(\frac{t - \tau}{a}\right). \tag{2}$$

$\psi$ is called the *mother wavelet*, $\tau \in \mathbb{R}$ is the *translation time*, $a \in \mathbb{R}_0^+$ is the *scale*, and $c(a) \sim a^m$ with $m \in \mathbb{Q}$. We can write the CWT as a convolution product,

$$S_x(\tau, a) = (\psi_a^\sharp \star x)(\tau), \tag{3}$$

where

$$\psi_a^\sharp(t) = c(a) \overline{\psi\left(\frac{-t}{a}\right)}, \tag{4}$$

in which $\overline{\cdot}$ denotes the complex conjugate. From the convolution theorem,

$$\widehat{S_x}(\omega, a) = \sqrt{2\pi} \widehat{\psi_a^\sharp}(\omega)\widehat{x}(\omega) = \sqrt{2\pi a}\, \overline{\widehat{\psi}(a\omega)}, \tag{5}$$

and $S_x(\tau, a)$ is then obtained by taking the inverse Fourier transform.

$|S_x(\tau, a)|^2$ gives the local power in the time-scale plane, and is called the *scalogram* by analogy with the *periodogram*.

### 2.2  The wavelet power spectrum

The wavelet power spectrum (WPS) of a continuous-time stochastic process $\{x(t)\}_{t \in \mathbb{R}}$ is defined by (see (Li and Oh, 2002) or (Maraun and Kurths, 2004)):

$$\mathrm{WPS}_x(\tau, a) = E\{|S_x(\tau, a)|^2\}, \tag{6}$$

where the expectation is taken over the samples of the stochastic process. A simple example is the WPS of a real-valued stationary white noise. Define $\{\eta(t)\}_{t \in \mathbb{R}}$ satisfying the following covariance property:

$$E\{\eta(t)\eta(t')\} = \sigma^2 \delta(t - t'). \tag{7}$$

Its WPS is then

$$\mathrm{WPS}_\eta(\tau, a) = a\, c(a)^2 ||\psi||^2 \sigma^2. \tag{8}$$



## 2.3 The Morlet wavelet as the mother wavelet

In this article, we choose the mother wavelet $\psi$ to be the Morlet wavelet (Grossmann and Morlet, 1984):

$$\psi(t) = \pi^{-1/4}\sigma_0^{-1/2}[\exp(i\omega_0 t) - \exp(-\omega_0^2\sigma_0^2/2)]\exp(-t^2/2\sigma_0^2), \tag{9}$$

This mother wavelet is a complex plane wave weighted by a Gaussian, to which is added a correction term to make

it *admissible*[2], i.e. satisfying $\int_{-\infty}^{+\infty}dt\,\psi(t) = 0$. This correction term is negligible[3] for $\sigma_0\omega_0 \geq 5.5$. If this inequality is satisfied, and with the variable change $a' = a/\omega_0$, the CWT with the Morlet wavelet is

$$S(\tau, a') = c(a')\int_{-\infty}^{+\infty}dt\,\exp\left(-\frac{i(t-\tau)}{a'}\right)\exp\left(-\frac{(t-\tau)^2}{2\sigma_0^2\omega_0^2 a'^2}\right)x(t), \tag{10}$$

where $c(a') \sim (a')^m$, $m \in \mathbb{Q}$, and holds all the multiplicative constants. Without loss of generality, we impose $\sigma_0 = 1$, and assume that

$\omega_0 \geq 5.5,$ $\qquad\qquad(11)$

is fulfilled in the following of this article. Therefore,

$$S(\tau, a) = c(a)\int_{-\infty}^{+\infty}dt\,\exp\left(-\frac{i(t-\tau)}{a}\right)\exp\left(-\frac{(t-\tau)^2}{2\omega_0^2 a^2}\right)x(t) = (\psi_a^\sharp \star x)(\tau), \tag{12}$$

where

$$\psi_a^\sharp(t) = c(a)\exp(it/a)\exp(-t^2/2\omega_0^2 a^2). \tag{13}$$

Under this form, interpreting Eq. (12) is straightforward: the CWT is the inner product between the signal $x$ and a Gaussian wave packet centered in $\tau = t$, of period $2\pi a$, and with a numerical support[4] of length $6\omega_0 a$. As the scale increases (resp. decreases), the support becomes wider (resp. narrower).

## 2.4 On the parameter $c(a)$

There are two common choices for $c(a)$ (Maraun and Kurths, 2004, Sect. 3)). The first one is $c(a)$ proportional to

$1/\sqrt{a}$,

$$c(a) \sim \frac{1}{\sqrt{a}}, \tag{14}$$

and gives a constant $\mathbf{L}^2$-norm for $\psi_{\tau,a}$, namely $||\psi_{\tau,a}|| = ||\psi||$. This implies that the wavelet power spectrum of a white noise is flat, as we can see in Eq. (8). The second choice is

$$c(a) \sim \frac{1}{a}, \tag{15}$$

---

[2]The admissibility criteria is required for $\psi$ to be a wavelet (Holschneider, 1995, p. 5).
[3]We have $\exp(-(5.5)^2) = 7.288.10^{-14}$ and $\exp(-(5.5)^2/2) = 2.700.10^{-07}$.
[4]The length of the support of the Gaussian may be approximated by six times its standard deviation.



and gives the same maximal power for sines of the same amplitude and with different frequencies. Indeed, from the Fourier transform of $\psi_a^{\sharp}$,

$$\widehat{\psi_a^{\sharp}}(\omega) = c(a)a\omega_0 \exp\left(-\frac{\omega_0^2(\omega a - 1)^2}{2}\right), \tag{16}$$

and applying Eq. (5), we must require $c(a)a$ to be constant to have the maxima of the scalogram of a sum sine waves

(all with the same amplitude and different frequencies) invariant with the scale.

### 2.5  The parameter $\omega_0$ and the time-frequency resolution

Parameter $\omega_0$ controls the time-frequency resolution, as it can be seen from the standard deviations of the Gaussian weights in $\psi_a^{\sharp}$, Eq. (13), and in its Fourier transform, Eq. (16). The standard deviations are equal to $\omega_0 a$ and $1/\omega_0 a$ respectively. Consequently, for a fixed scale, increasing (resp. decreasing) the value of $\omega_0$ will generate a CWT

with a better (resp. worse) frequency resolution and a worse (resp. better) time resolution. This property is of primary importance for the applications to time series, as illustrated in Sect. 7. Note that, for any time-frequency transform, there is always a trade-off between time and frequency localization. This property is often compared to the *Heisenberg uncertainty principle*. The Morlet wavelet exhibits the best trade-off, thanks to its Gaussian shape. We provide further details on this topic in appendix B.

### 2.6  Scale to period conversion

The Morlet wavelet is often used to detect the periodicities in a signal, and it is therefore suitable to convert scales $a$ into periods $T$ (Meyers et al., 1993). In practice, take a signal $x(t) = A\exp(i\omega t) = A\exp(i2\pi t/T)$. Its scalogram writes

$$|S(\tau, a)|^2 = 2\pi A c(a)^2 \omega_0^2 a^2 \exp(-\omega_0^2(\omega a - 1)^2), \tag{17}$$

and is independent of $\tau$. Scale to period conversion is performed with the value of the scale for which $|S(\tau, a)|^2$ is maximum (as a function of $a$). We find:

$$T = \begin{cases} 2\pi a & \text{if } c(a) \sim 1/a, \\ \frac{4\pi\omega_0 a}{\omega_0 + \sqrt{\omega_0^2 + 2}} & \text{if } c(a) \sim 1/\sqrt{a}. \end{cases} \tag{18}$$

For a fixed scale, and while $\omega_0 \geq 5.5$, difference between both never exceeds 2 %.

### 2.7  Reconstruction with the amplitude ridges

Reconstruction of a signal can be performed with the CWT along the *amplitude ridges*[5] (Carmona et al., 1997), which are the lines going through the maxima of the scalogram. Indeed, take the signal $x(t) = A\exp(i\omega t)$ and $c(a) \sim 1/a$.

---

[5]There also exist the *phase ridges*, defined in (Delprat et al., 1992), but we consider only the amplitude ridges in this study since they are easier to generalize to irregularly time series. A comparison of both the amplitude and phase ridges is found in (Lilly and Olhede, 2010).





Its scalogram is maximum at $a = 1/\omega$ (from Eq. (17)) and we can therefore easily recover the amplitude $A$ at each time $\tau$, going through the ridge $a(\tau) = 1/\omega$ in the scalogram, on which we have $|S(\tau, 1/\omega)| = \alpha A \;\; \forall \tau$, where $\alpha \in \mathbb{R}$ is a multiplicative constant. Jointly with the amplitude, the full signal $x(t)$ can be exactly recovered from the CWT along the ridge.

This can be extended to signals with slowly varying amplitude and phase, see (Delprat et al., 1992) and (Carmona et al., 1997), namely,

$$x(t) = A(t) \exp(i\phi(t)), \quad \text{s.t.} \quad \left| \frac{d\phi}{dt} \right| \gg \left| \frac{1}{A} \frac{dA}{dt} \right|, \tag{19}$$

for which the CWT taken along the ridge, i.e. at the maxima of its modulus, can approximately reconstruct $x(t)$. The inequality in Eq. (19), called *asymptoticity* condition, means that the instantaneous frequency inside the *wave packet*
must be much smaller than the frequency of the amplitude of the wave packet. The analysis can be further extended to a sum of asymptotic signals plus noise, and detected by multiple ridges (Carmona et al., 1999). When considering a real signal like $x(t) = A(t)\cos(\phi(t))$, we have to work with its *analytic* counterpart, which is built from the Fourier transform of $x$, $\widehat{x}$, for which we impose $\widehat{x}(\omega < 0) = 0$, and then take the inverse Fourier transform. Analyticity ensures that the phase and amplitude of a signal are uniquely determined. See (Lilly and Olhede, 2010) and the references
therein for more details. State of the art algorithms for ridge detection are developed in (Lilly and Olhede, 2010) and are available for use in the package jLab (https://github.com/jonathanlilly/jLab), in which the ridge-finding algorithm is general enough to be applied to various mother wavelets, like the Morlet wavelet.

## 2.8    Writing the scalogram under the formalism of orthogonal projections

Finally, we mention that the scalogram can be written under the formalism of orthonormal projections. Indeed,
defining

$$y_{\tau,a}(t) = \frac{\pi^{-1/4}}{\sqrt{\omega_0 a}} \exp\left( \frac{i(t-\tau)}{a} \right) \exp\left( -\frac{(t-\tau)^2}{2\omega_0^2 a^2} \right), \tag{20}$$

which has a unit norm, the scalogram can be formulated as

$$|S(\tau, a)|^2 = \gamma(a) |\langle y_{\tau,a} | x \rangle|^2 = \gamma(a) \left\| \frac{|y_{\tau,a}\rangle \langle y_{\tau,a}|}{\langle y_{\tau,a} | y_{\tau,a} \rangle} | x \rangle \right\|^2, \tag{21}$$

where $\gamma(a) = \alpha \;\; (\alpha \in \mathbb{R})$ if $c(a) \sim 1/\sqrt{a}$, or $\gamma(a) \sim 1/a$ if $c(a) \sim 1/a$.



## 3 The continuous wavelet transform of irregularly sampled time series

### 3.1 The model for the data

We consider the same model as in paper I:

$$|X\rangle = |\text{Trend}\rangle + E_{\tau,a}\cos(\omega|t\rangle + \phi_{\tau,a}) + |\text{Noise}\rangle$$

$$\qquad = |\text{Trend}\rangle + A_{\tau,a}|c_\omega\rangle + B_{\tau,a}|s_\omega\rangle + |\text{Noise}\rangle, \tag{22}$$

where $|X\rangle = [X_1,...,X_N]'$ and is real, $|t\rangle = [t_1,...,t_N]'$, $A_{\tau,a} = E_{\tau,a}\cos(\phi_{\tau,a})$, $B_{\tau,a} = -E_{\tau,a}\sin(\phi_{\tau,a})$, $E_{\tau,a}^2 = A_{\tau,a}^2 + B_{\tau,a}^2$, $|c_\omega\rangle = [\cos(\omega t_1),...,\cos(\omega t_N)]'$ and $|s_\omega\rangle = [\sin(\omega t_1),...,\sin(\omega t_N)]'$. We have added subscripts $(\tau,a)$ since all the subsequent analyses will be done in the time-scale plane. The trend is a polynomial of degree $m$,

$$|\text{Trend}\rangle = \sum_{k=0}^{m} \gamma_k |t^k\rangle, \tag{23}$$

and the background noise term, $|\text{Noise}\rangle$, is a zero-mean stationary Gaussian CARMA process sampled at the times of $|t\rangle$, as defined in Sect. 3.2 of paper I.

### 3.2 The scalogram

When applying the CWT to finite discrete time series, a choice for the discretization must be made. In the influential paper (Torrence and Compo, 1998), which deals with regularly sampled time series, the expression under the form of a convolution product in the Fourier space, Eq. (5), is conserved, and computed with the discrete Fourier transform (DFT) of the data. The CWT is then the inverse DFT of the convolution product. Unfortunately, we cannot extend the convolution theorem[6] to irregularly spaced time series and we cannot therefore follow the same computational procedure as in (Torrence and Compo, 1998). Alternatively, we can conserve the squared norm of the orthogonal projection, Eq. (21). The advantage of such a formalism is that it can be applied to irregularly sampled time series, as shown in paper I. Similarly to paper I, we work with cosines and sines instead of working with complex exponentials. A very little difference is observed between both choices. Based on the results of Sect. 2.8, our Morlet wavelet scalogram for irregularly sampled time series is therefore

$$||P_{\overline{\text{sp}}\{G_{\tau,a}\mathbf{c}_{\tau,\mathbf{a}}, G_{\tau,a}\mathbf{s}_{\tau,\mathbf{a}}\}}|X\rangle||^2, \tag{24}$$

where $G_{\tau,a}$ is a diagonal matrix with diagonal elements

$$G_{ii_{\tau,a}} = \exp\left(-\frac{(t_i - \tau)^2}{2\omega_0^2 a^2}\right) \quad \forall i \in \{1,...,N\}, \tag{25}$$

and

$$|c_{\tau,a}\rangle = \cos((|t\rangle - \tau)/a), \qquad |s_{\tau,a}\rangle = \sin((|t\rangle - \tau)/a), \tag{26}$$

---

[6]The convolution theorem for continuous-time functions is given in appendix A, and its counterpart for regularly sampled time series is given in (Mallat, 2009, p. 74).



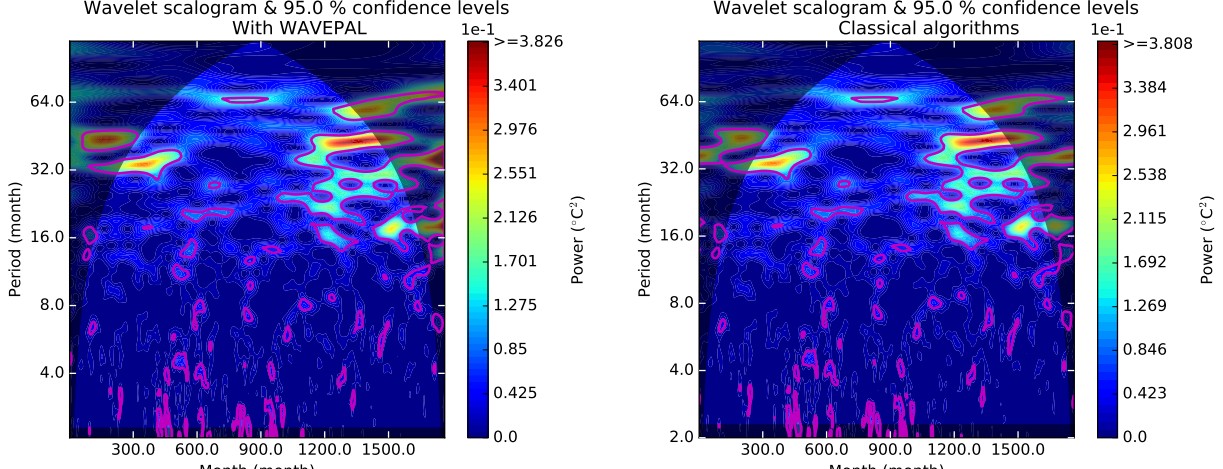

**Figure 1.** Comparison of the scalograms (with $\omega_0 = 15$) of regularly sampled time series, computed with WAVEPAL (left) or with the classical approach, based on (Torrence and Compo, 1998), (Maraun and Kurths, 2004) or (Cohen and Walden, 2010) (right). The time series is the NINO3 anomalies (https://www.esrl.noaa.gov/psd/gcos_wgsp/Timeseries/Nino3/). The analytical confidence levels, against a red noise background, are also drawn. The two lateral shaded areas are the half-cones of influence (see Sect. 3.7), and the bottom shaded area is the refinement of the Shannon-Nyquist exclusion zone (defined in Sect. 3.10). The bounds of the color scale are the extrema of the scalogram over the non-shaded area. Technical details about the computation of the scalogram and its confidence levels are given in Sect. 3 and 4. We see that the two scalograms are visually identical, except in the half-cones of influence where small discrepancies can occur.

are vectors of length $N$. We can impose $\tau = 0$ into the cosine and sine terms, since $\overline{\mathrm{sp}}\{G_{\tau,a}\mathbf{c}_{\tau,\mathbf{a}}, G_{\tau,a}\mathbf{s}_{\tau,\mathbf{a}}\}$ is invariant with respect to the variable $\tau$ appearing in the cosine and sine, and the scalogram becomes

$$||P_{\overline{\mathrm{sp}}\{G_{\tau,a}\mathbf{c}_{\mathbf{a}}, G_{\tau,a}\mathbf{s}_{\mathbf{a}}\}}|X\rangle||^2. \tag{27}$$

5  In the following, the notations $|G_{\tau,a}c_a\rangle$ or $G_{\tau,a}|c_a\rangle$ refer to the same vector. Our wavelet scalogram is similar to the tapered periodogram defined in Sect. 4.4 of paper I, and its properties and generalizations will therefore be similar as well. In particular, variables $a$ and $\tau$ are considered as continuous variables, similarly to the continuous frequency variable of paper I.

When the time series is regularly sampled, the scalogram, given by Eq. (27), is extremely close to what is obtained

10  with the traditional approach based on the convolution theorem, e.g. in (Torrence and Compo, 1998), (Maraun and Kurths, 2004) or (Cohen and Walden, 2010). This is illustrated in Fig. 1. Note that Eq. (27) reduces to the Lomb-Scargle periodogram, defined in Eq. (I,36), if the weight $G$ is set to unity.

Similarly to the Lomb-Scargle periodogram, we rescale $|G_{\tau,a}c_a\rangle$ and $|G_{\tau,a}s_a\rangle$ such that they are orthonormal. This



can be done by defining

$$|c_a^\sharp\rangle = \frac{\cos(|t\rangle/a - \beta_{\tau,a})}{\sqrt{\Sigma_{i=1}^N G_{ii_{\tau,a}}^2 \cos^2(t_i/a - \beta_{\tau,a})}}, \qquad |s_a^\sharp\rangle = \frac{\sin(|t\rangle/a - \beta_{\tau,a})}{\sqrt{\Sigma_{i=1}^N G_{ii_{\tau,a}}^2 \sin^2(t_i/a - \beta_{\tau,a})}}, \qquad (28)$$

where $\beta_{\tau,a}$ is the solution of

$$\tan(2\beta_{\tau,a}) = \frac{\Sigma_{i=1}^N G_{ii_{\tau,a}}^2 \sin(2t_i/a)}{\Sigma_{i=1}^N G_{ii_{\tau,a}}^2 \cos(2t_i/a)}. \qquad (29)$$

The scalogram is then

$$||P_{\overline{\text{sp}}\{G_{\tau,a}\mathbf{c_a}, G_{\tau,a}\mathbf{s_a}\}}|X\rangle||^2 = \langle G_{\tau,a}c_a^\sharp | X\rangle^2 + \langle G_{\tau,a}s_a^\sharp | X\rangle^2. \qquad (30)$$

### 3.3  Scalogram and trend

Analogously to Sect. 4.3 of paper I, we extend the scalogram to take into account the presence of a polynomial trend of degree $m$ in the data. Indeed, the scalogram defined in Sect. 3.2 applies well to data which can be modeled as

$|X\rangle = A_{\tau,a}|c_\omega\rangle + B_{\tau,a}|s_\omega\rangle + |\text{Noise}\rangle$. If we want to work with the full model, Eq. (22), holding a polynomial trend of degree $m$, we define a new scalogram as

$$||(P_{\overline{\text{sp}}\{\mathbf{t^0},\mathbf{t^1},...,\mathbf{t^m}, G_{\tau,a}\mathbf{c_a}, G_{\tau,a}\mathbf{s_a}\}} - P_{\overline{\text{sp}}\{\mathbf{t^0},\mathbf{t^1},...,\mathbf{t^m}\}})|X\rangle||^2, \qquad (31)$$

which is invariant with respect to the parameters of the trend. This is the analogue of Eq. (I,51).

### 3.4  Smoothing the scalogram

The scalogram suffers from the same inconsistency issue as the periodogram, in the sense that it remains very noisy whatever the number of data points we have at our disposal[7]. Smoothing techniques must therefore be applied, and we proceed like in paper I, extending the formulas used with regularly sampled time series. Note that the disadvantage of any smoothing procedure is that the resolution (in time, frequency or both, depending on the smoothing choice) is reduced. Consequently, there is always a trade-off between variance reduction and resolution.

Smoothing is traditionally performed by averaging the scalogram over neighboring points in the time-scale plane, either by averaging over times followed by averaging over scales (Torrence and Webster, 1999), (Grinsted et al., 2004), or simply by averaging over time (Cohen and Walden, 2010). In this work, we apply the latter technique because, even for very simple signals like $|X\rangle = \sin(\omega|t\rangle)$, the correlations in the scalogram between neighboring scales, for a fixed time, are highly irregular when the time series is irregularly sampled, unlike the correlations between

neighboring times, for a fixed scale, which are driven by the Gaussian shape of the wave packets $|G_{\tau,a}c_a\rangle$ and $|G_{\tau,a}s_a\rangle$. Smoothing over time must be carried out in accordance with the length of the support of the wave packets, which

---

[7]The scalogram often looks *smooth* because neighboring points in the time-frequency plane are strongly correlated, but it nevertheless remains inconsistent. See the discussion in (Maraun and Kurths, 2004, Sect. 4.2).




is proportional to the scale and to parameter $\omega_0$ (Eq. (25)). This choice also implies that the number of oscillations over which smoothing is performed is constant throughout the time-scale plane. This results from Eq. (26).

We adopt here the smoothing procedure of (Cohen and Walden, 2010) for which they derived analytical asymptotic results in the case of regularly sampled time series. The averaging window is a square window with a length

proportional to the scale. Our smoothed scalogram is

$$||P_{\text{smoothed}}(\tau,a)|X\rangle||^2 = \frac{1}{2\gamma\omega_0 a} \int_{\tau-\gamma\omega_0 a}^{\tau+\gamma\omega_0 a} \mathrm{d}\tau' ||(P_{\overline{\mathrm{sp}}\{\mathbf{t^0},\mathbf{t^1},\ldots,\mathbf{t^m},\mathbf{G}_{\tau',\mathbf{a}}\mathbf{c_a},\mathbf{G}_{\tau',\mathbf{a}}\mathbf{s_a}\}} - P_{\overline{\mathrm{sp}}\{\mathbf{t^0},\mathbf{t^1},\ldots,\mathbf{t^m}\}})|X\rangle||^2, \tag{32}$$

in which $\gamma$ is called the *smoothing coefficient*. Appendix D provides further details on the practical implementation of the bounds of integration.

### 3.5   The amplitude scalogram

### 3.5.1   Definition

We want to estimate the amplitude $E_{\tau,a} = \sqrt{A_{\tau,a}^2 + B_{\tau,a}^2}$ of our model, Eq. (22), at a given point $(\tau,a)$ of the time-scale plane. The estimation of $E_{\tau,a}^2$ is called the *amplitude scalogram* and is denoted by $\widehat{E}_{\tau,a}^2$. We start with a trendless signal and derive an approximate proportionality between the amplitude scalogram and the scalogram.

### 3.5.2   Trendless signal

Formula (I,111) is applied with the left-hand side term changed to encompass wavelet formalism. $\widehat{A}$ and $\widehat{B}$ are determined by projecting the data onto the tapered cosine and sine:

$$P_{\overline{\mathrm{sp}}\{\mathbf{G}_{\tau,\mathbf{a}}\mathbf{c_a},\mathbf{G}_{\tau,\mathbf{a}}\mathbf{s_a}\}}|X\rangle = \widehat{A}|c_\omega\rangle + \widehat{B}|s_\omega\rangle = V_{\omega_2}|\widehat{\Phi}\rangle, \tag{33}$$

where the taper $G_{\tau,a}$ is defined in Eq. (25),

$$V_{\omega_2} = \begin{pmatrix} | & | \\ |c_\omega\rangle & |s_\omega\rangle \\ | & | \end{pmatrix}, \text{ and } |\widehat{\Phi}\rangle = \begin{pmatrix} \widehat{A} \\ \widehat{B} \end{pmatrix}. \tag{34}$$

Conversion from the angular frequency $\omega$ to the scale $a$ is performed with the formula $\omega = 1/a$ (justification is given in Sect. 3.9). Using the same development as in Sect. 6.2.2 of paper I, we obtain

$$|\widehat{\Phi}_{\tau,a}\rangle = (V_{a_2}' G_{\tau,a} V_{a_2})^{-1} V_{a_2}' G_{\tau,a} |X\rangle. \tag{35}$$

The amplitude scalogram is then

$$\widehat{E}_{\tau,a}^2 = || |\widehat{\Phi_{\tau,a}}\rangle ||^2. \tag{36}$$




The approximations made in Sect. 6.2.2 of paper I are valid in this work, and applying Eq. (I,120) to our case gives an approximate proportionality between the scalogram and the amplitude scalogram, namely

$$\widehat{E}^2_{\tau,a} \approx \frac{2\mathrm{tr}(G^2_{\tau,a})}{\mathrm{tr}(G_{\tau,a})^2}||P_{\overline{\mathrm{sp}}\{G_{\tau,a}\mathbf{c_a},G_{\tau,a}\mathbf{s_a}\}}|X\rangle||^2. \tag{37}$$

Let us compare this equation with its continuous counterpart, Eq. (21), in which the weight must be $\gamma(a) \sim 1/a$ to
get an estimation of the local squared amplitude, as explained in Sect. 2.4. The comparison is made by analyzing the weight of the right-hand side term of Eq. (37) in the continuous limit:

$$\frac{1}{\overline{\Delta t}}\frac{2\mathrm{tr}(G^2_{\tau,a})}{\mathrm{tr}(G_{\tau,a})^2} \longrightarrow \frac{2\int_{-\infty}^{+\infty}\mathrm{d}t\exp\left(-\frac{(t-\tau)^2}{\omega_0^2 a^2}\right)}{\left(\int_{-\infty}^{+\infty}\mathrm{d}t\exp\left(-\frac{(t-\tau)^2}{2\omega_0^2 a^2}\right)\right)^2} = \frac{\sqrt{2}}{\omega_0 a}, \tag{38}$$

where $\overline{\Delta t}$ is the average time step. This is proportional to $1/a$ and it is therefore consistent with the continuous case.

### 3.5.3  Signal with a trend

Formula (I,121) is applied with the left-hand side term changed to encompass wavelet formalism:

$$P_{\overline{\mathrm{sp}}\{\mathbf{t^0},\mathbf{t^1},\ldots,\mathbf{t^m},G_{\tau,a}\mathbf{c_a},G_{\tau,a}\mathbf{s_a}\}}|X\rangle = \sum_{k=0}^{m}\widehat{\gamma}_k|t^k\rangle + \widehat{A}|c_\omega\rangle + \widehat{B}|s_\omega\rangle = V_{\omega_{m+3}}|\widehat{\Phi}\rangle, \tag{39}$$

where

$$V_{\omega_{m+3}} = \begin{pmatrix} | & & | & | & | \\ |t^0\rangle & \ldots & |t^m\rangle & |c_\omega\rangle & |s_\omega\rangle \\ | & & | & | & | \end{pmatrix}, \text{ and } |\widehat{\Phi}\rangle = \begin{pmatrix} \widehat{\gamma}_0 \\ \vdots \\ \widehat{\gamma}_m \\ \widehat{A} \\ \widehat{B} \end{pmatrix}. \tag{40}$$

Conversion from the angular frequency $\omega$ to the scale $a$ is performed with the formula $\omega = 1/a$ (justification is given
in Sect. 3.9). Using the same development as in Sect. 6.3 of paper I, we obtain

$$|\widehat{\Phi}_{\tau,a}\rangle = (W'_{\tau,a_{m+3}}V_{a_{m+3}})^{-1}W'_{\tau,a_{m+3}}|X\rangle, \tag{41}$$

where $W_{\tau,a_{m+3}}$ is identical to $V_{a_{m+3}}$ except in the last two columns, where the cosine and sine are tapered by $G_{\tau,a}$. This gives

$$\widehat{A}_{\tau,a} = \widehat{\Phi}_{\tau,a}(m+2), \qquad \widehat{B}_{\tau,a} = \widehat{\Phi}_{\tau,a}(m+3), \tag{42}$$

where $\widehat{\Phi}_{\tau,a}(m+2)$ and $\widehat{\Phi}_{\tau,a}(m+3)$ are the two last components of vector $|\widehat{\Phi}_{\tau,a}\rangle$. The amplitude scalogram is then

$$\widehat{E}^2_{\tau,a} = \widehat{A}^2_{\tau,a} + \widehat{B}^2_{\tau,a} \tag{43}$$




### 3.5.4 With smoothing

Like in paper I, estimating the amplitude is more robust against noise when a smoothing procedure is performed.
We apply to the squared amplitude, Eq. (43), the same kind of smoothing as for the scalogram, see Eq. (32), giving

$$\widehat{E}_{\tau,a}^2 = \frac{1}{2\gamma\omega_0 a} \int_{\tau-\gamma\omega_0 a}^{\tau+\gamma\omega_0 a} \mathrm{d}\tau'(\widehat{\Phi}_{\tau,a}(m+2)^2 + \widehat{\Phi}_{\tau,a}(m+3)^2).$$ (44)

Appendix D provides further details on the practical implementation of the bounds of integration.

### 3.6 The weighted smoothed scalogram

The weighted smoothed scalogram is the analogue of the *weighted WOSA periodogram*, defined in Sect. 7 of paper
I, and its objectives are the same, i.e. to keep the advantages of both the amplitude scalogram and the scalogram,
namely:

– Provide an estimation of the squared amplitude of a signal, locally in the time-frequency plane, by weighting
the scalogram like in Eq. (37).

– Conserve the advantage of the formalism of orthogonal projections, in order to avoid the matrix inversions for
the computation of the amplitude scalogram. See e.g. Eq. (44), relying on Eq. (41) which requires a matrix
inversion.

The last item is useful for building confidence levels when performing a test of significance (see Sect. 4). The
disadvantage of weighting the smoothed scalogram is that it does not provide anymore a flat *pseudo-wavelet spectrum*
for a white noise signal (see Sect. 4.2), analogously to its frequency counterpart (see Sect. 7 of paper I). The weighted
smoothed scalogram is derived from Eq. (32), in which the integrand is weighted by the right-hand side weight of
Eq. (37), namely,

$$||P_{\mathrm{smoothed}}(\tau,a)|X\rangle||^2 = \frac{1}{2\gamma\omega_0 a} \int_{\tau-\gamma\omega_0 a}^{\tau+\gamma\omega_0 a} \mathrm{d}\tau' \frac{2\mathrm{tr}(G_{\tau',a}^2)}{\mathrm{tr}(G_{\tau',a})^2} ||(P_{\overline{\mathrm{sp}}\{\mathbf{t^0},\mathbf{t^1},...,\mathbf{t^m},\mathbf{G_{\tau',a}c_a},\mathbf{G_{\tau',a}s_a}\}} - P_{\overline{\mathrm{sp}}\{\mathbf{t^0},\mathbf{t^1},...,\mathbf{t^m}\}})|X\rangle||^2$$ (45)

Appendix D provides further details on the practical implementation of the bounds of integration. We recommend
the use of the weighted smoothed scalogram in most time-frequency analyses under irregular sampling.

### 3.7 Cone of influence

When the wave packets $|G_{\tau,a}c_a\rangle$ and $|G_{\tau,a}s_a\rangle$ intersect the borders of the time series, a part of their support can stand
after the last point of the time series, or before the first point of the time series. Consequently, one has to remove
two half-cones from the area under analysis. From Eq. (25), the support of the wave packets is approximately equal
to $2\beta\omega_0 a$, so that the excluded areas are given by

$$\{\tau,a\} \quad \mathrm{s.t.} \quad |\tau-t_1| \leq \beta\omega_0 a \quad \mathrm{and} \quad |\tau-t_N| \leq \beta\omega_0 a,$$ (46)




with $\beta = 3$ (conservative choice) or $\beta = \sqrt{2}$ (choice in (Torrence and Compo, 1998)). We recommend the conservative choice. When smoothing is performed, Eq. (46) becomes

$$\{\tau, a\} \quad \text{s.t.} \quad |\tau - t_1| \leq (\beta + \gamma)\omega_0 a \quad \text{and} \quad |\tau - t_N| \leq (\beta + \gamma)\omega_0 a, \tag{47}$$

where $\gamma$ is controlling the smoothing length, see Eq. (32), (44) and (45). This has another implication: the maximal

scale available by the analysis is

$$a_{\max} = \frac{t_N - t_1}{2(\beta + \gamma)\omega_0}. \tag{48}$$

### 3.8 Aliasing and Shannon-Nyquist exclusion zone (SNEZ)

When probing the irregularly sampled time series with the wavelet packet, it may happen that the period of the oscillation inside the packet, $2\pi a$, is too low compared to the local time step in the time series, therefore causing

aliasing issues according to the Shannon-Nyquist theorem, locally in the time-scale plane. As stated in paper I, this issue also happens with the WOSA periodogram. We adapt formulas (I,66), (I,67) and (I,68) and define the local time step by

$$\overline{\Delta t}_{\tau, a} = \max\{\overline{\Delta t}_{G_{\tau, a}}, \overline{\Delta t}_{H_{\tau, a}}\}, \tag{49}$$

where

$$\overline{\Delta t}_{G_{\tau, a}} = \frac{\sum_{k=1}^{N} G_{\tau, a_{k,k}} \Delta t_{c_k}}{\operatorname{tr}(G_{\tau, a})}, \qquad \overline{\Delta t}_{H_{\tau, a}} = \frac{\sum_{k=1}^{N-1} H_{\tau, a_{k,k}} \Delta t_k}{\operatorname{tr}(H_{\tau, a})}, \tag{50}$$

$$\Delta t_k = t_{k+1} - t_k, \ \Delta t_{c_k} = \frac{t_{k+1} - t_{k-1}}{2} \ \forall k \in \{2, ... N-1\}, \ \Delta t_{c_1} = t_2 - t_1, \ \Delta t_{c_N} = t_N - t_{N-1}, \tag{51}$$

and

$$H_{\tau, a} = \text{diagonal matrix with } H_{\tau, a_{k,k}} = \exp\left(-\frac{\left(\frac{t_k + t_{k+1}}{2} - \tau\right)^2}{2\omega_0^2 a^2}\right), \ k \in \{1, ..., N-1\}. \tag{52}$$

We then apply the Shannon-Nyquist theorem to this local time step, namely

Compute the scalogram at $(\tau, a)$ if $a \geq a_{\text{SNEZ}}(\tau)$, \hfill (53)

where

$a_{\text{SNEZ}}(\tau)$ is the largest solution of $a = \dfrac{\overline{\Delta t}_{\tau, a}}{\pi}$. \hfill (54)

We call *Shannon-Nyquist exclusion zone* (SNEZ) the area in the scalogram that does not satisfy Eq. (53) and which

is therefore delimited by $a_{\text{SNEZ}}$. Note that matrix $H_{\tau, a}$ is similar to matrix $G_{\tau, a}$, defined in Eq. (25), but with elements



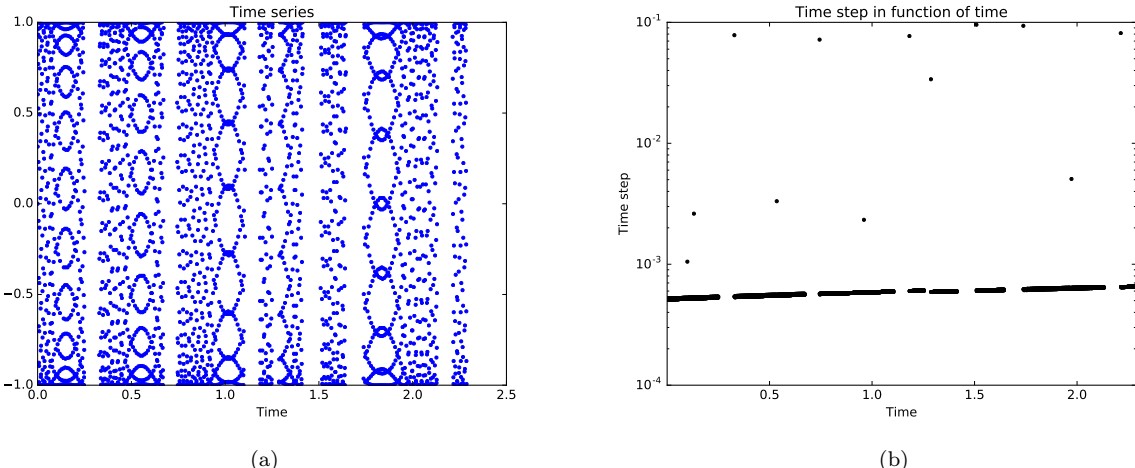

**Figure 2.** (a) The time series $|X\rangle = \sin(2\pi|t\rangle/0.01)$, and (b) its time step. The time vector $|t\rangle$ is taken from a real paleoclimate time series (Giosan, 2017)

taken at $(t_k + t_{k+1})/2$ instead of $t_k$. Quantity $\overline{\Delta t}_{\tau,a}$ is equal to the maximum between the average weighted time step and the average weighted central time step.

We now justify formula (49) with an example. Consider the function $X(t) = \sin(2\pi t/0.01)$, sampled on an irregular grid. This is drawn on Fig. 2a. The time step is represented in Fig. 2b. These two figures show that the time series

exhibits intervals where it is more or less regularly sampled, separated by large gaps. The weighted (unsmoothed) scalogram is drawn on Fig. 3a. We remind that the weighted scalogram is supposed to estimate the local squared amplitude in the time-frequency plane. Since $X(t)$ has an amplitude equal to 1, we expect that the maximal power of the scalogram be equal to 1, along a scale corresponding to the period of $x$, for all $\tau$. Because of the large gaps in the time series, extended regions corrupted by aliasing occur in Fig. 3a, resulting in a maximal power for the scalogram

which is much greater than 1. Figures 3b, 3c and 3d present the weighted scalogram corrected by the SNEZ. In Fig. 3b the SNEZ is computed with $\overline{\Delta t}_{\tau,a} = \overline{\Delta t}_{G_{\tau,a}}$. We observe that it does a good job at rejecting the areas where aliasing occur, although it is desirable that the black areas peak at higher scales. In Fig. 3c, the SNEZ is computed with $\overline{\Delta t}_{\tau,a} = \overline{\Delta t}_{H_{\tau,a}}$. We observe that most of the aliasing-related areas are rejected, although we wish wider black areas. Finally, the SNEZ computed with $\overline{\Delta t}_{\tau,a} = \max\{\overline{\Delta t}_{G_{\tau,a}}, \overline{\Delta t}_{H_{\tau,a}}\}$ is drawn on Fig. 3d and we observe that it does a very

satisfactory job at rejecting the areas where aliasing occur.

The SNEZ is applied to all the analysis tools defined above. When smoothing is to be applied, it is performed on the areas outside of the SNEZ, since the scalogram is not computed in the SNEZ. In the neighborhood of the SNEZ, adjustments of the smoothing procedure are therefore necessary, as explained in appendix D.



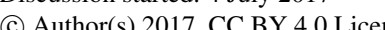

**Figure 3.** Weigthed (unsmoothed) scalogram of the time series presented on Fig. 2a. (a) No correction for aliasing. (b) Corrected with $\overline{\Delta t}_{\tau,a} = \overline{\Delta t}_{G_{\tau,a}}$. (c) Corrected with $\overline{\Delta t}_{\tau,a} = \overline{\Delta t}_{H_{\tau,a}}$. (d) Corrected with $\overline{\Delta t}_{\tau,a} = \max\{\overline{\Delta t}_{G_{\tau,a}}, \overline{\Delta t}_{H_{\tau,a}}\}$.





### 3.9 From the scale to the period

Scale to period conversion is performed in the continuous limit, with Eq. (18). The first case of Eq. (18), with $c(a) \sim 1/a$, corresponds to estimators of the *amplitude*, and is then used for scale to period conversion with the amplitude scalogram (all the formulas of Sect. 3.5) and for the weighted smoothed scalogram, Eq. (45). The second

case of Eq. (18), with $c(a) \sim 1/\sqrt{a}$, is used for scale to period conversion with the unweighted scalogram, that is the formulas appearing in Sect. 3.2, 3.3 and 3.4.

### 3.10 Refining the Shannon-Nyquist exclusion zone

As illustrated on Fig. 4, the Shannon-Nyquist exclusion zone may not to be sufficient to avoid all the patches due to aliasing, because of the correlations between neighboring scales in the scalogram. We therefore extend the

Shannon-Nyquist exclusion zone by considering the continuous limit case for the simple periodic signal $x(t) = \exp(i2\pi t/T_{\mathrm{SNEZ}}(\tau))$, where $T_{\mathrm{SNEZ}}(\tau)$ is the period at the border of the SNEZ, determined by Eq. (54) and (18). Its scalogram is given in Eq. (17). In order to make the correspondence with all the above formulas, three cases are considered:

1. $c(a) \sim 1/a$: In this case, we have $|S(\tau,a)|^2 \sim \exp(-\omega_0^2(2\pi a/T_{\mathrm{SNEZ}}(\tau) - 1)^2)$, and the standard deviation for the

scale is then $\sigma_{a,1}(\tau) = T_{\mathrm{SNEZ}}(\tau)/2\sqrt{2}\pi\omega_0$. The border of the *extended* Shannon-Nyquist exclusion zone at time $\tau$ is therefore at scale $a_{\mathrm{SNEZ}}(\tau) + \beta\sigma_{a,1}(\tau)$, where $\beta$ is a coefficient estimating the half support of Gaussian shaped functions (it is defined in Sect. 3.7).

2. $c(a) \sim 1/a$ and work with $|S(\tau,a)|$: In this case, we have $|S(\tau,a)| \sim \exp(-\omega_0^2(2\pi a/T_{\mathrm{SNEZ}}(\tau) - 1)^2/2)$, and the standard deviation for the scale is then $\sigma_{a,2}(\tau) = T_{\mathrm{SNEZ}}(\tau)/2\pi\omega_0$. The border of the extended Shannon-Nyquist

exclusion zone at time $\tau$ is therefore at scale $a_{\mathrm{SNEZ}}(\tau) + \beta\sigma_{a,2}(\tau)$.

3. $c(a) \sim 1/\sqrt{a}$: In this case, we have $|S(\tau,a)|^2 \sim a\exp(-\omega_0^2(2\pi a/T_{\mathrm{SNEZ}}(\tau) - 1)^2)$. We know from Eq. (18) that the scalogram is maximum at the scale $a_{\max}(\tau) = T_{\mathrm{SNEZ}}(\tau)(\omega_0 + \sqrt{\omega_0^2 + 2})/4\pi\omega_0$. The pseudo-standard deviation is computed such that
   $a\exp(-\omega_0^2(2\pi a/T_{\mathrm{SNEZ}}(\tau) - 1)^2)$ decreases from its maximum by the same percentage as in case 1, namely,

$\beta\sigma_{a,3}(\tau)$ is equal to the largest of the two solutions of

$$a\exp(-\omega_0^2(2\pi a/T_{\mathrm{SNEZ}}(\tau) - 1)^2) = a_{\max}\exp(-\omega_0^2(2\pi a_{\max}(\tau)/T_{\mathrm{SNEZ}}(\tau) - 1)^2)\exp(-\beta^2/2),$$

in which the unknown is $a$. The border of the extended Shannon-Nyquist exclusion zone at time $\tau$ is therefore at scale $a = a_{\mathrm{SNEZ}}(\tau) + \beta\sigma_{a,3}(\tau)$.

Case 1 is used with formulas giving the squared amplitude $\widehat{E}_{\tau,a}^2$ in Sect. 3.5 and with the weighted smoothed scalogram,

Eq. (45). The (unsquared) amplitude $\widehat{E}_{\tau,a}$ can also be of interest, and case 2 is therefore used. Case 3 is used with formulas arising in Sect. 3.2, 3.3 and 3.4. Finally, note that the refinement of the SNEZ is performed after the



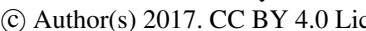

**Figure 4.** Scalogram of the time series $|X\rangle = \sin(2\pi|t\rangle/10)$, where $|t\rangle$ has a piecewise constant time step. From $t = 0$ to $t = 200$, $\Delta t = 4$. From $t = 200$ to $t = 400$, $\Delta t = 3$. From $t = 400$ to $t = 600$, $\Delta t = 2$. (a) Weighted (unsmoothed) scalogram. The black area is the SNEZ. (b) Same as (a) with the refinement of the SNEZ, which is the shaded area on the top of the SNEZ. (c) Amplitude scalogram (unsmoothed). The black area is the SNEZ. (d) Same as (c) with the refinement of the SNEZ, which is the shaded area on the top of the SNEZ. In the 4 panels, the bounds of the color scale are the extrema of the scalogram over the non-shaded area. Thanks to the refinement of the SNEZ, the upper bound of the color scale is close to 1, which is the value of the (squared) amplitude of the signal $|X\rangle$.

smoothing procedure, because an extension of the SNEZ may result from the smoothing, as explained in appendix D.





## 3.11 Discretizing $\tau$ and $a$

With regularly sampled data, the discretized variable $\tau$ is usually equal to $|t\rangle$, like in (Torrence and Compo, 1998), or a subset of $|t\rangle$ with regularly spaced elements. For irregularly spaced time series, we opt for the same type of grid as in the regularly sampled case, i.e. a linear regular grid, namely

$$\tau_k = \tau_0 + k\Delta\tau, \quad k \in \{0,...,K\}, \quad \text{with } \tau_0 \geq t_1 \text{ and } \tau_K \leq t_N. \tag{55}$$

The scales are commonly discretized as fractional powers of two (Torrence and Compo, 1998), namely

$$a_j = a_{\min}2^{j\delta j}, \quad j = 0,...,J, \tag{56}$$

where

$$J = \log_2(a_{\max}/a_{\min})/\delta j. \tag{57}$$

$a_{\min}$ is the minimum over $\tau$ of $a_{\mathrm{SNEZ}}$ (defined in Eq. (54)), and $a_{\max}$ is defined in Eq. (48). Discretization as a power law comes from the geometry of the wavelet transform, and is justified in appendix C.

The integrals in Eq. (32), (44) and (45) are discretized with the rectangle method. In particular, the discretized integrals from Eq. (32) and (45) allows to write these formulas as finite-size matrices. To this end, we apply a Gram-Schmidt orthonormalization to the orthogonal projections, like in Eq. (I,61). This gives

$$||P_{\mathrm{smoothed}}(\tau,a)|X\rangle||^2 = \langle X|M_{2_{\tau,a}}M'_{2_{\tau,a}}|X\rangle, \tag{58}$$

which is the analogue of Eq. (I,62). $M_{2_{\tau,a}}$ is a matrix of size $(N, 2n_{\mathrm{col}}(\tau,a))$, $n_{\mathrm{col}}(\tau,a) >= 1$, where $n_{\mathrm{col}}$ is a non-trivial function depending on the scale and on the closeness of $(\tau,a)$ with the SNEZ and with edges of the time-frequency plane.

## 4 Significance testing with the scalogram

### 4.1 Hypothesis testing

We test for the presence of periodic components, locally in the time-frequency plane. Significance testing is mathematically expressed as a hypothesis testing. Taking our model, Eq. (22), the null hypothesis is

$$H_0 : A_{\tau,a} = B_{\tau,a} = 0. \tag{59}$$

Therefore, $|X\rangle = |\mathrm{Trend}\rangle + |\mathrm{Noise}\rangle$. The alternative hypothesis is

$$H_1 : A_{\tau,a} \text{ and } B_{\tau,a} \text{ are not both zero.} \tag{60}$$

The decision of accepting or rejecting the null hypothesis is based on the scalogram (Eq. (45)), independently for each couple $(\tau,a)$ (this is called *pointwise testing*). Concretely, for each couple $(\tau,a)$, we compute the distribution





of the scalogram under the null hypothesis, and then see if the *data* scalogram at $(\tau, a)$ is above or below a given percentile of that distribution. The percentile is called *level of confidence*. If the data scalogram is above the $X^{th}$ percentile of the reference distribution, we reject the null hypothesis with X % of confidence. The *level of significance* is equal to $(100 - X)$ %, e.g. a 95 % confidence level is equivalent to a 5 % significance level.

To perform significance testing, we thus need

   1. to estimate the parameters of the process under the null hypothesis. This is studied in Sect. 5.2 of paper I.

   2. to estimate the distribution of the scalogram under the null hypothesis. This is studied in Sect. 4.2 below.

Finally, we mention that, for regularly sampled time series, the pointwise significance test can be supplemented with an *areawise* significance test, which takes into account the correlations between neighboring points in the time-
frequency plane. This is introduced in (Maraun and Kurths, 2004) and studied in detail in (Maraun et al., 2007). Applying this method to irregularly sampled series is way beyond the scope of this work, since the correlations between neighboring points in the time-frequency plane are highly irregular.

### 4.2  Estimation of the distribution of the scalogram under the null hypothesis

The results obtained for the periodogram in Sect. 5.3 of paper I are valid for the scalogram, with minor changes that
we detail below.

   1. Monte-Carlo approach: The same procedure as in paper I is applied to the (weighted) smoothed scalogram, Eq. (32) or (45). We can thus estimate the confidence levels for the (weighted) smoothed scalogram taking into account the uncertainty on the parameters of the background noise.

   2. Analytical approach (with a unique set of CARMA parameters):

– Theorem 1 of paper I can be applied to the (weighted) smoothed scalogram, as follows.

         **Theorem 1.** *The (weighted) smoothed scalogram, defined in Eq. (58), under the null hypothesis (59), is*[8]

$$||P_{smoothed}(\tau, a)|X\rangle||^2 \overset{d}{=} \sum_{k=1}^{2n_{col}(\tau, a)} \lambda_k(\tau, a)\chi_1^2, \tag{61}$$

         *where $|X\rangle = \sum_{k=0}^{m} \gamma_k |t^k\rangle + K|Z\rangle$ and $K$ is the CARMA matrix defined in Eq. (I,15) or (I,34). The $\chi_1^2$*
*distributions are iid, and $\lambda_1(\tau, a)$, ..., $\lambda_{2n_{col}(\tau, a)}(\tau, a)$ are the eigenvalues of $M'_{2\tau, a} KK' M_{2\tau, a}$ and are non-negative. Matrix $M_{2\tau, a}$ is defined in Eq. (58).*

---

[8]The symbol $\overset{d}{=}$ means "is equal in distribution".



- The *pseudo-wavelet power spectrum*, $\widehat{\mathrm{WPS}}$, is the analogue of the pseudo-spectrum defined in Eq. (I,85). It is defined as the expected value of the (weighted) smoothed scalogram distribution, namely

$$\widehat{\mathrm{WPS}}(\tau,a) = \sum_{k=1}^{2n_{\mathrm{col}}(\tau,a)} \lambda_k(\tau,a) = \mathrm{tr}(M'_{2_{\tau,a}}KK'M_{2_{\tau,a}}). \tag{62}$$

- For a Gaussian white noise background with variance $\sigma^2$, the unweighted pseudo-wavelet power spectrum is flat, and is equal to $2\sigma^2$, for all $(\tau,a)$. Moreover, if the scalogram is not smoothed, it is exactly chi-square-distributed with 2 degrees of freedom:

$$||(P_{\overline{\mathrm{sp}}\{\mathbf{t^0},\mathbf{t^1},\ldots,\mathbf{t^m},G_{\tau,a}\mathbf{c_a},G_{\tau,a}\mathbf{s_a}\}} - P_{\overline{\mathrm{sp}}\{\mathbf{t^0},\mathbf{t^1},\ldots,\mathbf{t^m}\}})\sigma|Z\rangle||^2 \stackrel{d}{=} \sigma^2\chi_2^2, \tag{63}$$

where $|Z\rangle$ is a standard Gaussian white noise.

- The variance of the distribution of the (weighted) smoothed scalogram at $(\tau,a)$ is equal to $2||M'_{2_{\tau,a}}KK'M_{2_{\tau,a}}||_F^2$, where $||\cdot||_F$ is the Frobenius norm.

- We approximate the linear combination of the independent chi-square distributions, appearing in Eq. (61), by a gamma-polynomial distribution conserving its first $d$ moments, based on the theory developed in (Provost et al., 2009). The formulas are given in Sect. 5.3.3 of paper I.

We observe, however, that the convergence of the percentiles (as the number of conserved moments grows) strongly depends on the smoothing coefficient $\gamma$, defined in Eq. (32) and (45). As a general rule, the larger is $\gamma$, the faster is the convergence. Moreover, it turns out that the gamma-polynomial approximation becomes numerically unstable at large numbers of conserved moments, because the matrix in Eq. (I,93) becomes singular. Consequently, for relatively small values of $\gamma$, convergence cannot be numerically guaranteed. This is illustrated on Fig. 5. In such cases, a simple 2-moment approximation is therefore a reasonable choice since it is always numerically stable, it is much quicker than with higher numbers of conserved moments from a computational point of view, and it provides a satisfactory approximation.

3. A comparison between the computing times of the Monte-Carlo approach and the analytical approach is presented in appendix G.

## 5 Filtering with the amplitude scalogram

### 5.1 Band filtering

From Sect. 3.5.3, Eq. (42) gives $\widehat{A}_{\tau,a}$ and $\widehat{B}_{\tau,a}$. We can therefore reconstruct the signal $\widehat{A}_{\tau,a}|c_a\rangle + \widehat{B}_{\tau,a}|s_a\rangle$ over the whole time-scale plane, i.e. for all $(\tau,a)$. Band filtering is performed by averaging the reconstructed signal between scales $a_{j_{\mathrm{min}}}$ and $a_{j_{\mathrm{max}}}$, namely

$$X_{\mathrm{filt}}(\tau) = \frac{1}{j_{\mathrm{max}} - j_{\mathrm{min}} + 1} \sum_{j=j_{\mathrm{min}}}^{j_{\mathrm{max}}} \widehat{A}_{\tau,a_j}|c_{a_j}\rangle + \widehat{B}_{\tau,a_j}|s_{a_j}\rangle, \tag{64}$$



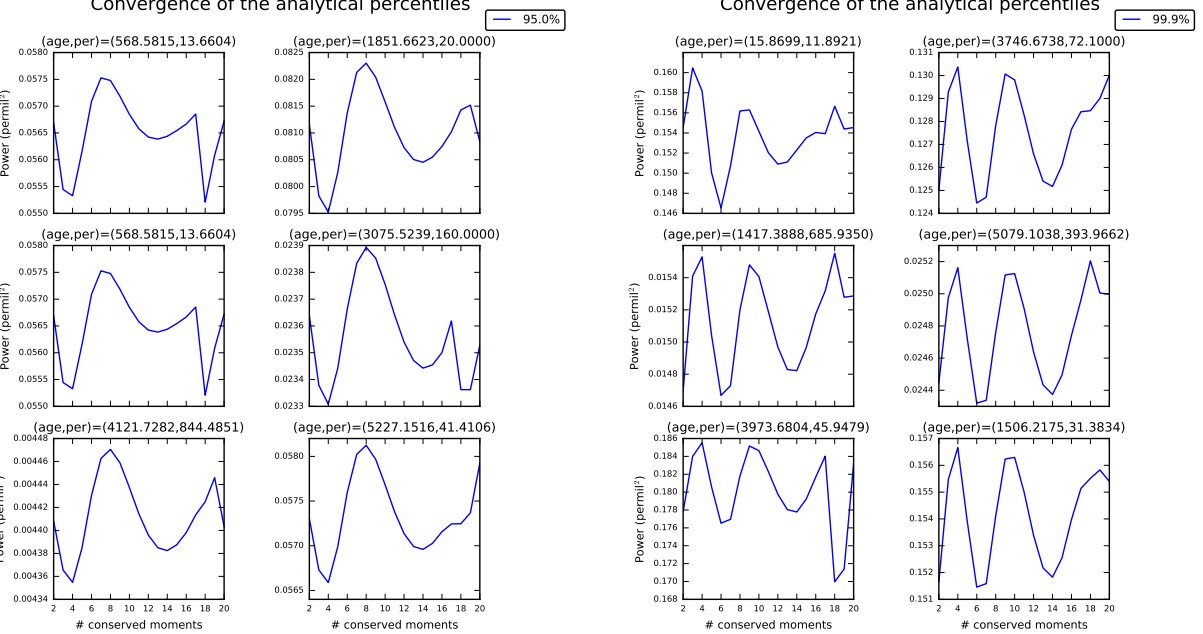

**Figure 5.** Analytical confidence levels in function of the number of conserved moments, at six particular couples $(\tau, a)$, for the scalogram of the time series presented in Sect. 7. Parameter $\gamma$ is equal to 0.5. Left panel: $95^{\text{th}}$ percentile. Right panel: $99.9^{\text{th}}$ percentile. Slow convergence as well as numerical instabilites (spurious peaks) at high numbers of conserved moments are observed. Convergence cannot therefore be numerically guaranted.

where the discretized scale is defined in Eq. (56). Such filtering is a generalization of the *scale-averaged wavelet power* of (Torrence and Compo, 1998) which deals with trendless regularly sampled signals. Note that we use the formulas for which there is no smoothing. Indeed, the smoothing procedure in Eq. (44) does not give access to $\widehat{A}_{\tau,a}$ and $\widehat{B}_{\tau,a}$ (only the sum of their squared values is available). An example of band filtering is shown in Fig. 9 and 10.

## 5.2 Ridge filtering

Consider a signal $|X\rangle = E\cos(\omega|t\rangle + \phi)$. We can easily reconstruct the signal from the estimated amplitudes $\widehat{A}_{\tau,a}$ and $\widehat{B}_{\tau,a}$, given by Eq. (42), taken at the maximum of the scalogram, in this case at $a = 1/\omega$. More generally, we can reconstruct more complex signals relying on the theory of the *amplitude ridges*, developed for the continuous case (Sect. 2.7), and which can approximately be applied to irregularly spaced time series. An example of ridge filtering is shown in Fig. 9 and 10.




## 6 The global scalogram

Analogously to the *global wavelet spectrum* of (Torrence and Compo, 1998) for trendless regularly sampled time series, we define here the global scalogram as the scalogram averaged over time. Technically, it is nothing but the smoothed scalogram (Eq. (32), (44) or (45)) with integration over the whole interval of the analysis time $\tau$. We can write the discretized global scalogram under a similar matrix form as in Eq. (58), and find the confidence levels according to Sect. 4. Compared to the periodogram defined in paper I, which has a fixed bandwidth, the global scalogram has a varying bandwidth with the frequency. From Fig. C1 of appendix C, we deduce that the global scalogram exhibits a frequency resolution that gets better when the frequency decreases. Examples of global scalograms are given in Sect. 7.

## 7 Application on paleoceanographic data

### 7.1 Preliminary analysis

The time series we use to illustrate the theoretical results is the benthic foraminiferal $\delta^{18}O$ record from (Jian et al., 2003) that holds 608 data points with distinct ages and covers the last 6 million years. The choice of a CARMA(1,0) process as the background stationary noise, as well as the choice of $m = 7$ for the degree of the polynomial trend, are justified in Sect. 9 of paper I, in which the same data set is used as an example of frequency analysis. The time series, its trend and its time step are drawn on Fig. 6. We remind that the time series is not detrended before computing the scalogram of the data, but it is detrended before estimating the confidence levels.

### 7.2 Time-frequency analysis

The weighted smoothed scalogram (Sect. 3.6) and its 95 % analytical and MCMC confidence levels are presented on Fig. 7 with parameter $\omega_0 = 5.5$, and on Fig. 8 with parameter $\omega_0 = 15$. As explained in Sect. 2, increasing $\omega_0$ results in a better frequency resolution and a worse time resolution. In our example, the scalogram with $\omega_0 = 15$ exhibits more clearly the period band around 40 kyr and the changes in amplitude along that band.

The parameters are: $\gamma = 0.5$ (smoothing coefficient), 2 is the number of conserved moments in the gamma-polynomial approximation (see the discussion in Sect. 4.2), a fixed-length smoothing per scale (see appendix D), $\beta = 3$ (half-support of a standard Gaussian function $\exp(-x^2/2)$), and $\delta j = 0.05$ (coefficient for the scale resolution).

### 7.3 Filtering

As explained in Sect. 5, band and ridge filtering are performed on the unsmoothed amplitude scalogram. This is illustrated on Fig. 9, with a filtering band in the interval $[35, 45]$ kyr and with the ridges. From the whole set of the ridges, we select those in the band $[35, 45]$ kyr, in order to make a comparison with the band filtering. Band and ridge filtered signals are shown in Fig. 10. We can see that the amplitude modulations in Fig. 10a and 10b are consistent.




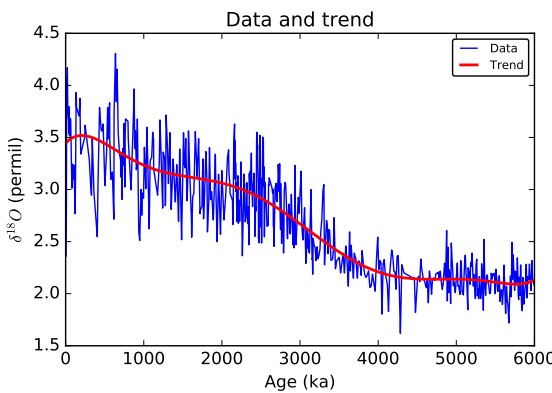
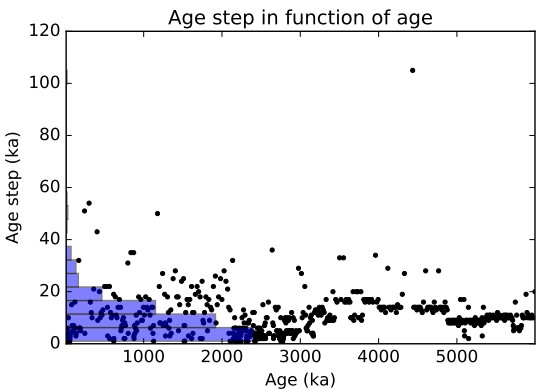

**Figure 6.** (a) The time series and its $7^{th}$ degree polynomial trend. (b) The age step, $[t_k - t_{k-1}]\ \forall k \in 2,...,N$, and its distribution.

Compared to band filtering, the ridge filtering method has the advantage of representing the signal for which the amplitude scalogram is locally maximal, and also allows to reconstruct the time-varying amplitude of the filtered signal (in red on Fig. 10b). The drawback is that it rarely delivers a continuous reconstruction with climate data.

## 8   WAVEPAL Python package

WAVEPAL is a package, written in Python 2.X, that performs frequency and time-frequency analyses of irregularly sampled time series, significance testing against a stationary Gaussian CARMA(p,q) process, and filtering. Frequency analysis is based on the theory developed in paper I, and time-frequency analysis relies on the theory developed in this article. It is available at https://github.com/guillaumelenoir/WAVEPAL.

## 9   Conclusions

We defined the scalogram as an extension of the generalized Lomb-Scargle periodogram developed in paper I. This analysis tool is well-suited for irregularly sampled time series which can be modeled as a locally periodic component in the time-frequency plane, plus a polynomial trend, plus a Gaussian CARMA stochastic process. In the particular case of trendless regularly sampled times series, we shown that the unsmoothed scalogram gives the same results as with the traditional algorithms such as in (Torrence and Compo, 1998). A smoothing procedure, by averaging over neighboring points in time, was then applied to the scalogram in order to reduce its variance. Besides, we derived estimators of the amplitude of the locally periodic component, based on the general results of paper I, and proposed an approximate proportionality between the scalogram and the squared amplitude. The latter result is at the basis of the weighted smoothed scalogram, which is the analysis tool that we recommend for most time-frequency analyses. We then shown that local aliasing issues may occur in the analysis tools previously derived, implying the delimitation



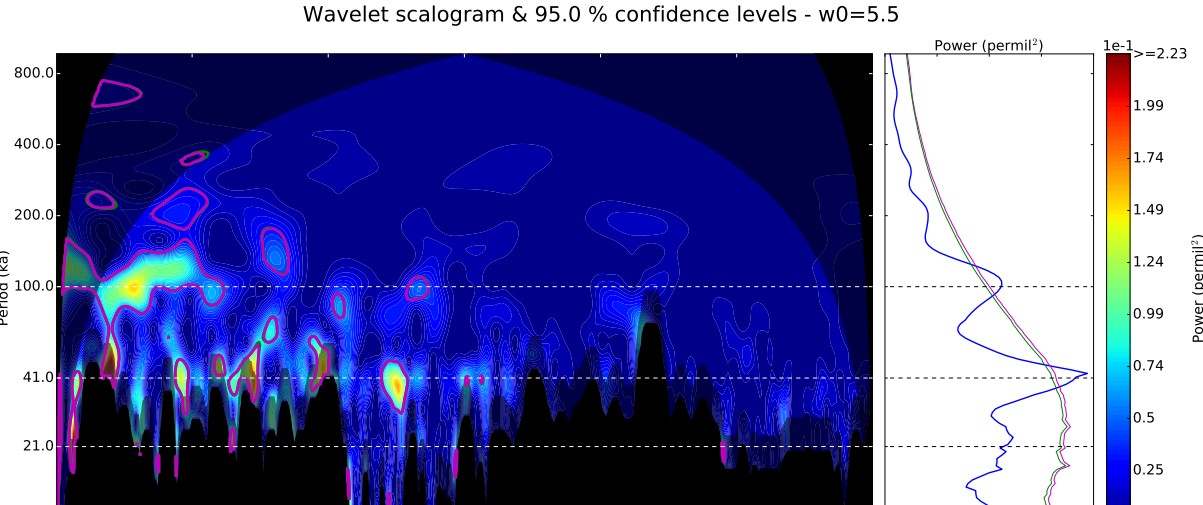

**Figure 7.** Weighted smoothed scalogram (left) and its global scalogram (right) with $\omega_0 = 5.5$. The 95 % analytical confidence levels (green) and 95 % MCMC confidence levels (magenta), against a red noise background, are also drawn. Note that the green and magenta contours are almost superposed. The two lateral shaded areas are the half-cones of influence, the bottom black area is the SNEZ, and the shaded area above the SNEZ is the refinement of the SNEZ. There are also two lateral black areas, where the scalogram is not computed, because of the fixed-length smoothing per scale. The bounds of the color scale are the extrema of the scalogram over the non-shaded area. As we work with the weighted scalogram, the power is an estimation of the local squared amplitude. Dashed lines at usual paleoclimate periods are also drawn.

of a forbidden area for the analyses, called the Shannon-Nyquist exclusion zone. Moreover, a test of significance for the scalogram was designed, similarly to its counterpart for the frequency analysis developed in paper I. Finally, the classical filtering procedures, namely band and ridge filtering, were made available for use with our operator of the estimated amplitude.

5 *Code availability.* The Python code generating the figures of this article is available in a supplementary material.

## Appendix A: Fourier analysis of functions

$\mathbf{L}^2(\mathbb{R})$ is the space of measurable functions on $\mathbb{R}$ with finite energy:

$$||f||^2_{\mathbf{L}^2} = \int_{-\infty}^{+\infty} \mathrm{d}t |f(t)|^2 < \infty. \tag{A1}$$



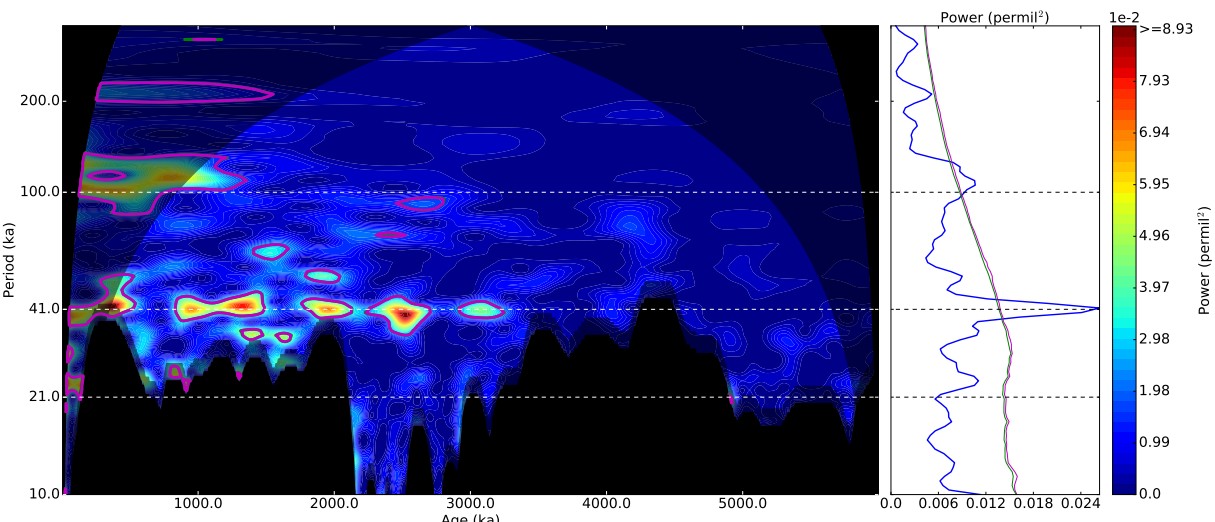

**Figure 8.** Weighted smoothed scalogram (left) and its global scalogram (right) with $\omega_0 = 15$. The 95 % analytical confidence levels (green) and 95 % MCMC confidence levels (magenta), against a red noise background, are also drawn. Note that the green and magenta contours are almost superposed.

This defines the squared norm for such functions, that we denote simply by $||f||^2$ in Sect. 2. We provide the $\mathbf{L^2}$ space with the usual inner product:

$$\langle f \,|\, g \rangle_{\mathbf{L^2}} = \int\limits_{-\infty}^{+\infty} \mathrm{d}t \, \overline{f(t)} g(t), \tag{A2}$$

which makes it a Hilbert space. $\langle f \,|\, g \rangle_{\mathbf{L^2}}$ is denoted by $\langle f \,|\, g \rangle$ in Sect. 2.

5 The Fourier transform of $f \in \mathbf{L^2}(\mathbb{R})$ is defined by

$$\widehat{f}(\omega) = \frac{1}{\sqrt{2\pi}} \int\limits_{-\infty}^{+\infty} \mathrm{d}t \, f(t) \exp(-i\omega t), \tag{A3}$$

which is the $\mathbf{L^2}$-inner product between $f(t)$ and $\exp(i\omega t)$. The inverse Fourier transform is

$$f(t) = \frac{1}{\sqrt{2\pi}} \int\limits_{-\infty}^{+\infty} \mathrm{d}\omega \, \widehat{f}(\omega) \exp(i\omega t). \tag{A4}$$

Some properties of the Fourier transform are listed below:

10     – Parseval-Plancherel identities: $\langle f \,|\, g \rangle = \langle \widehat{f} \,|\, \widehat{g} \rangle$ and $||f||^2 = ||\widehat{f}||^2$.

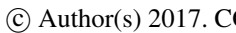



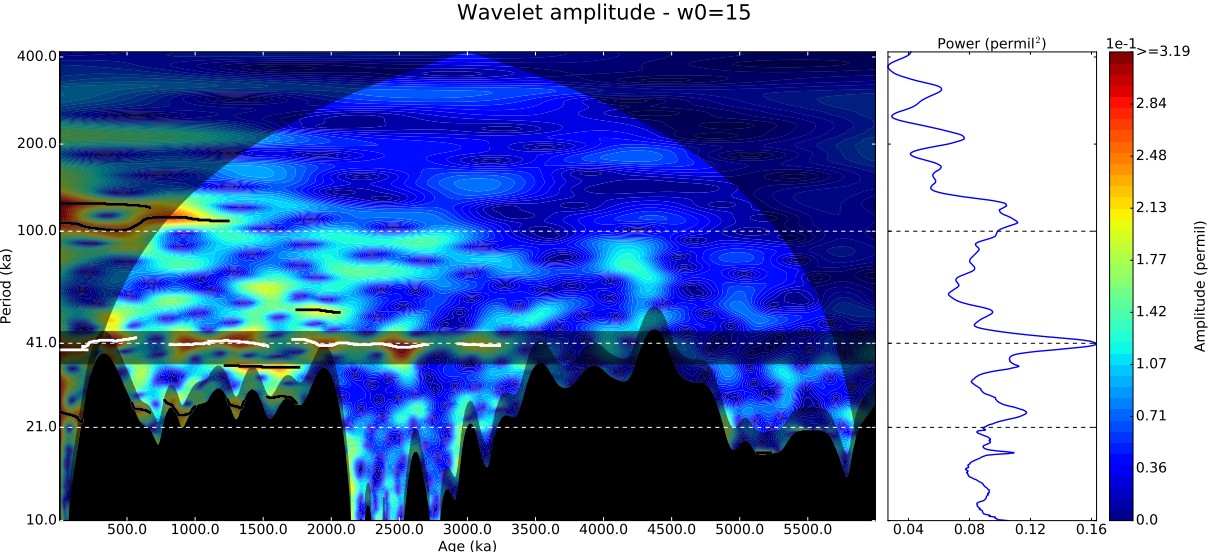

**Figure 9.** The unsmoothed estimated amplitude (which is the square root of the amplitude scalogram, Eq. (43)), jointly with the filtering band in the interval [35,45] kyr (shaded). Black and white curves are the ridges. They go through the local maxima of the amplitude scalogram. The white ones are the ridges in the band [35,45] kyr. Parameters are $\omega_0 = 15$, $\beta = 3$ and $\delta j = 0.01$.

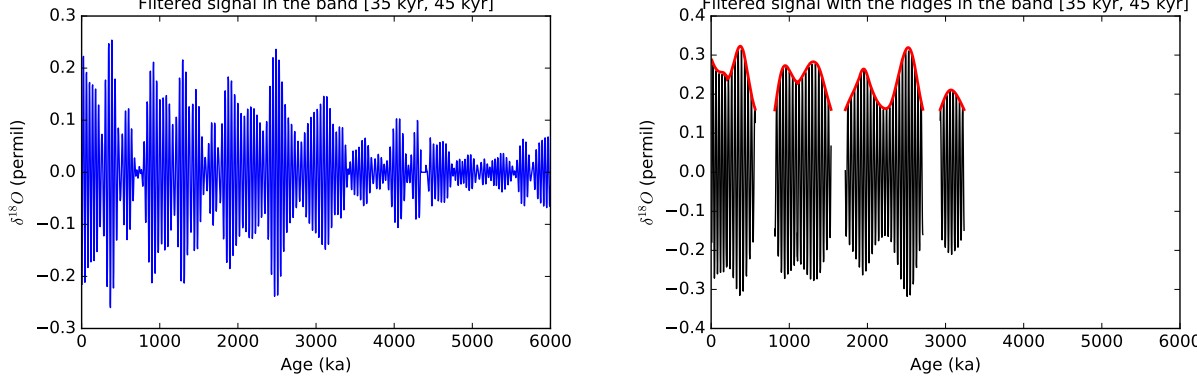

**Figure 10.** Filtered signal in the band [35,45] kyr. Left: Band filtering. Right: Ridge filtering. The red curve is the amplitude of the filtered signal, which is only available with ridge filtering.





- Convolution theorem: $[f \star g]\widehat{\ }(\omega) = \sqrt{2\pi}\widehat{f}(\omega)\widehat{g}(\omega)$, where the convolution product between $f$ and $g$ is $(f \star g)(t) = \int_{-\infty}^{+\infty} dt' f(t-t')g(t')$.

- Translation-modulation: The Fourier transform of $f(t-b)$ is $\exp(-i\omega b)\widehat{f}(\omega)$.

- Dilation: The Fourier transform of $f(at)$, $a \neq 0$, is $\frac{1}{|a|}\widehat{f}(\omega/a)$.

## 5 Appendix B: Heisenberg uncertainty for the Morlet wavelet

Heisenberg uncertainty theorem states that the temporal variance and the frequency variance of a function $f \in \mathbf{L}^2(\mathbb{R})$ satisfy

$$\sigma_t^2 \sigma_\omega^2 \geq \frac{1}{4}, \tag{B1}$$

where

$$10 \quad \sigma_t^2 = \frac{1}{\sqrt{2\pi}||f||^2} \int\limits_{-\infty}^{+\infty} dt\,(t-u)^2|f(t)|^2, \tag{B2}$$

and

$$\sigma_\omega^2 = \frac{1}{\sqrt{2\pi}||f||^2} \int\limits_{-\infty}^{+\infty} dt\,(\omega-\xi)^2|\widehat{f}(\omega)|^2. \tag{B3}$$

$\mu$ and $\xi$ are the average time and average frequency and are defined with the same densities as for the variances. For the Morlet wavelet, the densities are $|\psi_a^\sharp(t)|^2$ (from Eq. (13)) and $|\widehat{\psi_a^\sharp}(\omega)|^2$ (from Eq. (16)), up to a normalizing 15 multiplicative factor. As they are Gaussian functions, their variances are trivial, and we have

$$\sigma_t^2 \sigma_\omega^2 = \frac{\omega_0^2 a^2}{2} \frac{1}{2\omega_0^2 a^2} = \frac{1}{4}. \tag{B4}$$

This is equal to the lower bound of Heisenberg inequality, as expected for Gaussian functions. See (Mallat, 2009, p. 43) for additional details. It is in that sense that the Morlet wavelet is said to be ideally localized.

## Appendix C: Heisenberg boxes and scale discretization

### 20 C1 Time-frequency resolution and Heisenberg boxes

We saw in appendix B that the standard deviations of the continuous-time density $|\psi_a^\sharp(t)|^2$ and continuous-frequency density $|\widehat{\psi_a^\sharp}(\omega)|^2$ are $\sigma_t = \omega_0 a/\sqrt{2}$ and $\sigma_\omega = 1/\sqrt{2}\omega_0 a$ respectively. Moreover, the center angular frequency of $|\widehat{\psi_a^\sharp}(\omega)|^2$ is $\omega = 1/a$. With all these coefficients and Eq. (3) and (5), we can draw rectangles, called *Heisenberg boxes* (Mallat, 2009, p. 109), in the time-frequency plane indicating the energy spread around each couple $(\tau, \omega)$, or equivalently, 25 the time-frequency resolution at each couple $(\tau, \omega)$. This is illustrated in Fig. C1. Note that their area is equal to $\sigma_t \sigma_\omega = 1/2$ and is therefore constant.



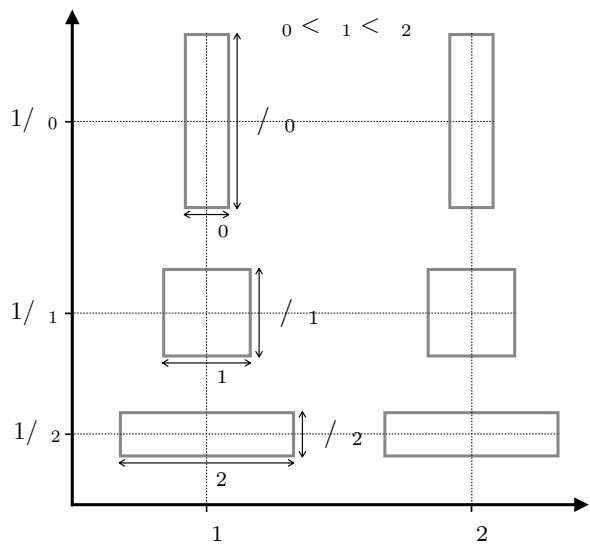

**Figure C1.** Heisenberg boxes for the Morlet wavelet, with $\alpha = \omega_0/\sqrt{2}$ and $\beta = 1/\sqrt{2}\omega_0$.

## C2    Scale discretization

Scale discretization is naturally based on the geometry of the Heisenberg boxes. We can, for example, require that the frequency component of the center of mass of the box corresponding to scale $a_j$ be at the frequency of the border of the box corresponding to scale $a_{j+1}$. This is illustrated on Fig. C2. We obtain

$$\frac{1}{a_j} = \frac{1}{a_{j+1}} + \frac{\beta}{2a_{j+1}}, \tag{C1}$$

where $\beta$ is defined in Fig. C1, giving

$$a_{j+1} = \left(\frac{2+\beta}{2}\right) a_j, \tag{C2}$$

and by recurrence,

$$a_{j+1} = \left(\frac{2+\beta}{2}\right)^j a_0. \tag{C3}$$

Multiplying $\beta$ by a positive factor, $\gamma$, allows to control the density of the discretized scales. With variable change $\delta j = \log_2[(2+\beta\gamma)/2]$, we obtain

$$a_{j+1} = 2^{j\delta j}a_0, \quad \delta j > 0, \quad \forall j \in \{0,...,J\}. \tag{C4}$$



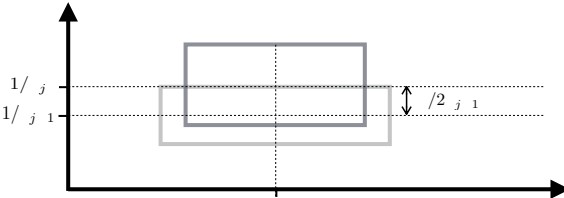

**Figure C2.** Example of rule for the discretization of scales taking into account the geometry of the Heisenberg boxes. $\beta = 1/\sqrt{2}\omega_0 > 0$.

## Appendix D: Smoothing the (amplitude) scalogram: Technical details

In the formulas of the smoothed (amplitude) scalogram, Eq. (32), (44) and (45), integration is in principle performed over the interval $[\tau - \gamma\omega_0 a, \tau + \gamma\omega_0 a]$. When this interval intersects the edges of the time-frequency plane or the SNEZ, we are no longer able to integrate over the full interval. Two choices are then possible:

1. Keep the length of integration equal to $2\gamma\omega_0 a$, and therefore exclude from the analysis some areas of the time-frequency plane. This results in two excluded zones at the time borders of the scalogram and in an extension of the SNEZ.

   2. Shorten the interval of integration in order to not exclude from the analysis any extra region of the time-frequency plane.

Both options are available in WAVEPAL and we recommend the first one, in order to keep a consistent degree of smoothing at each point $(\tau, a)$ in the time-scale plane.

## Appendix E: On Foster formulas for the Morlet-CWT of irregularly sampled time series

### E1    Introduction

In this appendix, we derive and comment about the formulas published in (Foster, 1996c), and based on developments
published in (Foster, 1996a) and (Foster, 1996b). Foster theory restricts to the case of the unsmoothed scalogram applied to signals with an additive Gaussian white noise and a piecewise trend for which the shape is the envelope of the Morlet wavelet. It also defines something similar to our amplitude scalogram and generalizes the F-periodogram of paper I. We show that some of its formulas can be deduced from our general theory[9]. Foster formulas are available for use in a Fortran code provided by the American Association of Variable Star Observers (AAVSO), see
(https://www.aavso.org/sites/default/files/software/wwz.tar.gz).

---

[9]See (Foster, 1996c) for the original derivation of the formulas, which is rather different from our approach.





### E2 Foster approximation and weighted inner products

Let us start with the approximation made in (Foster, 1996b) and used in (Foster, 1996c). Define $U$ as equal to a full rank real matrix, whose columns are the vectors generating the vector space on which we project the data vector $|X\rangle$, the latter belonging to $\mathbb{R}^N$. Define $G$ as equal to a real diagonal square matrix of size N with positive elements.

Foster approximation (Foster, 1996b, Eq. (7.9)) writes[10]

$$U'G^2U \approx \frac{\text{tr}(G^2)}{\text{tr}(G)}U'GU. \tag{E1}$$

Note that, when U is a 2-column matrix holding a cosine vector and a sine vector, the above approximation can also be obtained from Eq. (I,119). The orthogonal projection on the span of $GU$ thus becomes

$$P_{\overline{\text{sp}}\{\mathbf{GU}\}} = GU(U'G^2U)^{-1}U'G$$

$$\approx \frac{\text{tr}(G)}{\text{tr}(G^2)}GU(U'GU)^{-1}U'G, \tag{E2}$$

and, for any pair of vectors $|Y\rangle$ and $|W\rangle$ in $\mathbb{R}^N$, we have

$$\langle Y|P_{\overline{\text{sp}}\{\mathbf{GU}\}}|W\rangle \approx N_{\text{eff}}\frac{\langle Y|GU}{\text{tr}(G)}(U'GU)^{-1}\text{tr}(G)\frac{U'G|W\rangle}{\text{tr}(G)}, \tag{E3}$$

where $N_{\text{eff}} = \frac{\text{tr}(G)^2}{\text{tr}(G^2)}$ is defined in (Foster, 1996c, Eq. (7.7)) and called the *effective number of data points*. We can actually rewrite the right-hand side of Eq. (E3) as $N_{\text{eff}}\langle Y|P_{\overline{\text{sp}}\{\mathbf{U}\}}|W\rangle_{\text{Weighted}}$, where the weighted inner product is

defined by:

$$\langle Y\,|\,W\rangle_{\text{Weighted}} = \frac{\langle Y|G|W\rangle}{\text{tr}(G)}, \tag{E4}$$

for any $|Y\rangle$ and $|W\rangle$ in $\mathbb{R}^N$. $\langle\cdot|\cdot\rangle_{\text{Weighted}}$ satisfies to the requirements of an inner product since the elements of $G$ are positive, see (Brockwell and Davis, 1991, p. 43). Foster theory is developed in a vector space provided with this weighted inner product.

### E3 WWT

Now, we derive Foster scalogram from our theory. The diagonal elements of the weight matrix $G$ are (Foster, 1996c, Eq. (5-3)):

$$G_{kk_{\tau,\omega}} = \exp(-c\omega^2(t_k - \tau)^2). \tag{E5}$$

Correspondence with our weight matrix, defined in Eq. (25), is performed with the variable changes $\omega = 1/a$ and

$c = 1/2\omega_0^2$. Next, consider the formula of the unsmoothed scalogram, Eq. (31), with $a = 1/\omega$, and transformed to accommodate for a trend given by $|G_{\tau,\omega}t^0\rangle$. This results in

$$||P_{\overline{\text{sp}}\{\mathbf{G}_{\tau,\omega}\mathbf{t^0},\mathbf{G}_{\tau,\omega}\mathbf{c}_\omega,\mathbf{G}_{\tau,\omega}\mathbf{s}_\omega\}}|X\rangle||^2 - ||P_{\overline{\text{sp}}\{\mathbf{G}_{\tau,\omega}\mathbf{t^0}\}}|X\rangle||^2. \tag{E6}$$

---

[10]In (Foster, 1996b), the author works with tensor notations, so that the equivalence is not direct.





We then make use of the approximation of Eq. (E2) with $U = [|t^0\rangle \, |c_\omega\rangle \, |s_\omega\rangle]$ for the first projection and $U = |t^0\rangle$ for the second projection, resulting in the following formula:

$$N_{\text{eff}} \left( ||P_{\overline{\text{sp}}\{\mathbf{t^0}, \mathbf{c_\omega}, \mathbf{s_\omega}\}} |X\rangle ||^2_{\text{Weighted}} - ||P_{\overline{\text{sp}}\{\mathbf{t^0}\}} |X\rangle ||^2_{\text{Weighted}} \right), \tag{E7}$$

for which we now work in a vector space provided with the weighted inner product. If $|X\rangle$ is a zero-mean Gaussian white noise, formula (E6) follows exactly a chi-square distribution with 2 degrees of freedom multiplied by the variance of the white noise, namely $\sigma^2 \chi^2_2$. Consequently, under the null hypothesis that the process is a Gaussian white noise, the following expression

$$\text{WWT} = \frac{N_{\text{eff}}}{2\sigma^2} \left( ||P_{\overline{\text{sp}}\{\mathbf{t^0}, \mathbf{c_\omega}, \mathbf{s_\omega}\}} |X\rangle ||^2_{\text{Weighted}} - ||P_{\overline{\text{sp}}\{\mathbf{t^0}\}} |X\rangle ||^2_{\text{Weighted}} \right), \tag{E8}$$

approximately follows a chi-square distribution with 2 degrees of freedom and expected value 1. Formula (E8) is rigorously the same[11] as the *weighted wavelet transform* (WWT) of (Foster, 1996c), in which the author estimates $\sigma^2$ as

$$\widehat{\sigma}^2 = \frac{N_{\text{eff}}}{N_{\text{eff}} - 1} \left( \langle X | X \rangle_{\text{Weighted}} - \langle t^0 | X \rangle^2_{\text{Weighted}} \right). \tag{E9}$$

Significance testing against a Gaussian white noise can be therefore be performed with the WWT.

Below, we comment on the WWT and make a comparison with our formulas.

– The WWT is built on the assumption that the time series holds a Gaussian-shaped trend centered at the probed translation time $\tau$, the support of which varying with the probed frequency. This is equivalent to a constant trend in the vector space provided with the weighted inner product. This contrasts with our choice for the trend, Eq. (23), which is independent of the analysis function.

– The WWT under the null hypothesis is only *approximately* chi-square-distributed, compared to formula (E6) which is *exactly* chi-squared-distributed.

– The estimation of the variance of the white noise, $\widehat{\sigma}^2$, which is part of the WWT formula, depends on the sampling. However, two samples of a white noise are uncorrelated whatever is the time step separating them, and the estimation of its variance should thus be independent of the sampling, like in Sect. 5.2.2 of paper I.

– To our point of view, working with weighted inner products, approximations like in Eq. (E1), and complicated tensor notations (see (Foster, 1996b)) does not bring a simple and unified view of the problematic.

---

[11]Note that Eq. (5-10) of (Foster, 1996c), which is a prerequisite for the formula of the WWT, is probably erroneous, making unclear the correspondence with our Eq. (E8). However, the formula given here in Eq. (E8) is strictly the same as the WWT encoded in (https://www.aavso.org/sites/default/files/software/wwz.tar.gz).





### E4   WWA

The *weighted wavelet amplitude* (WWA), defined in (Foster, 1996c, Eq. (5-14)), is similar to our amplitude scalogram defined in Eq. (43). The former is obtained from the latter taking the trend to be $|G_{\tau,\omega}t^0\rangle$, where $G_{\tau,\omega}$ is defined in Sect. E3. For practical applications, we note that computing the inverse of a matrix is needed for the computation of

the WWA (this is also the case for our amplitude scalogram). But Foster theory lacks of an in-depth consideration of aliasing issues, and the WWA at some points of the time-frequency plane may be numerically infinite due to the occurrence of singular matrices caused by aliasing.

### E5   WWZ

Under the null hypothesis that the data $|X\rangle$ is a Gaussian white noise, its squared norm in the vector space provided

with the weighted inner product is approximately chi-square-distributed with $N_{\text{eff}}$ degrees of freedom, as this follows from the 2-moments approximation of Sect. 5.3.3 of paper I, from which formula (I,91) is applied to matrix $G$. Consequently, under the null hypothesis, the following formula

$$\frac{(N_{\text{eff}}-3)\left[||P_{\overline{\text{sp}}\{\mathbf{t^0},\mathbf{c}_\omega,\mathbf{s}_\omega\}}|X\rangle||^2_{\text{Weighted}} - ||P_{\overline{\text{sp}}\{\mathbf{t^0}\}}|X\rangle||^2_{\text{Weighted}}\right]}{2\left[||X||^2_{\text{Weighted}} - ||P_{\overline{\text{sp}}\{\mathbf{t^0},\mathbf{c}_\omega,\mathbf{s}_\omega\}}|X\rangle||^2_{\text{Weighted}}\right]}, \tag{E10}$$

is approximately equal to the Fisher-Snedecor distribution with 2 and $N_{\text{eff}}-3$ degrees of freedom. Formula (E10)

is defined in (Foster, 1996c, Eq. (5-12)) and called the *weighted wavelet Z-transform* (WWZ). It generalizes the F-periodogram that we defined in Sect. 5.4 of paper I.

### Appendix F: Warning about interpolating the time series

This appendix compares the scalograms and their confidence levels in the case of interpolated and non-interpolated time series. The time series we consider is the $\delta^{18}O$ signal from the GISP2 ice core (Grootes and Stuiver, 1997),

for which the first 11 kyr are removed, in order to facilitate the detrending procedure. The time series is drawn in Fig. F1b and F1c, and its time step is given in Fig. F1a. The interpolated time series is built on a time grid with $\Delta t = 30$ yr (this is is the smallest time step of the raw time series), see Fig. F1b, or $\Delta t = 300$ yr, see Fig. F1c. The (unsmoothed) scalograms with $\omega_0 = 15$ of the raw and interpolated time series are shown in Fig. F2, jointly with the 95 % analytical confidence levels against a red noise. We observe that confidence testing is strongly dependent on the

interpolation procedure. This is because the parameters of the red noise are badly estimated when the time series is interpolated. Consequently, in general, we cannot rely on interpolated time series to perform significance testing. In particular, we draw the attention on the geological *stacks*, such as in (Lisiecki and Raymo, 2005) or (Huybers, 2007), which are composed of multiple interpolated time series and averaged together. Significance testing or analysis of the background noise for such time series may therefore be strongly biased.

Finally, we observe that, in this example, the power of scalogram the data is weakly affected by the interpolation.





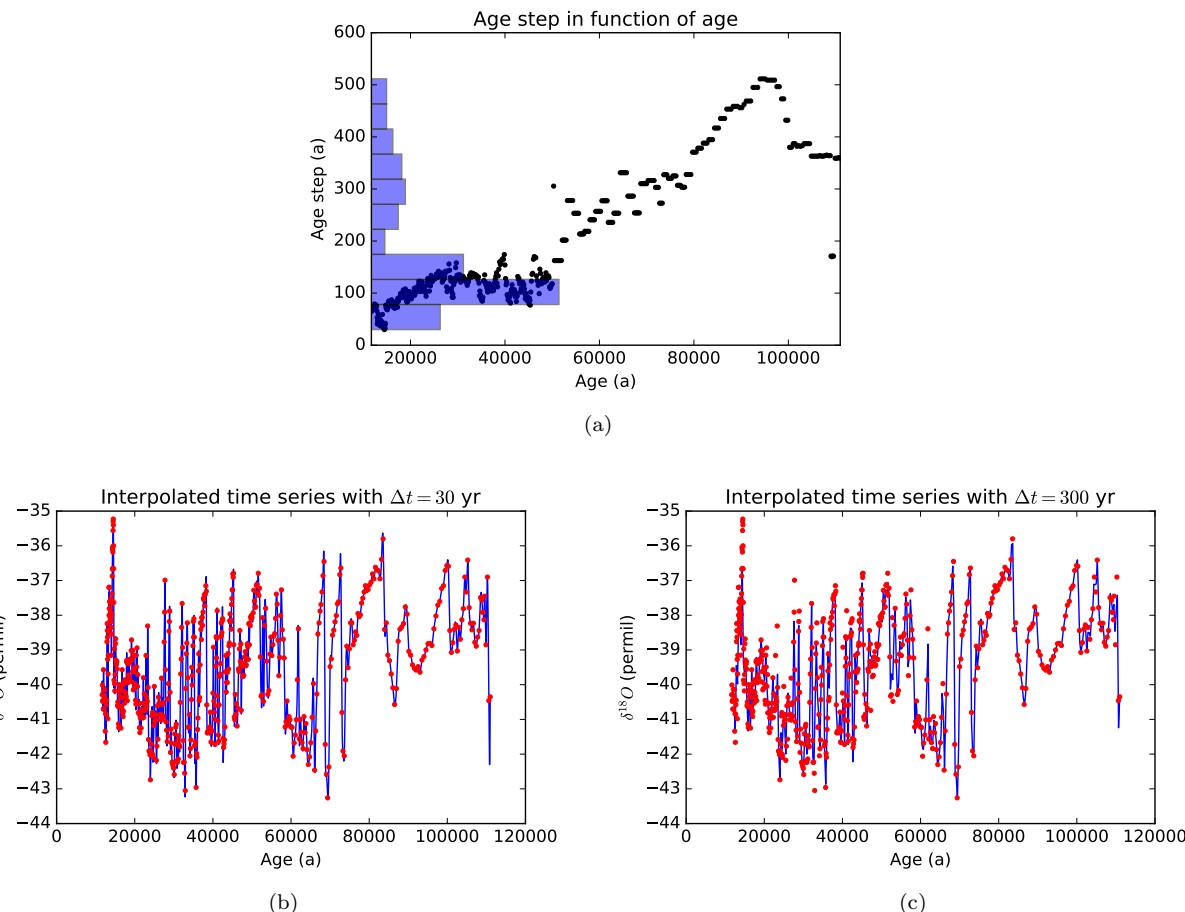

**Figure F1.** $\delta^{18}O$ signal from the GISP2 ice core (Grootes and Stuiver, 1997), for which the first 11 kyr are removed. (a) Time step. (b) The raw (red dots) and interpolated (blue line) time series with $\Delta t = 30$ yr. (c) The raw (red dots) and interpolated (blue line) time series with $\Delta t = 300$ yr.

## Appendix G: Computing time: Analytical versus Monte-Carlo significance levels

A comparison between the computing times, for generating the scalogram, with the analytical and with the MCMC confidence levels, based on the hypothesis of a red noise background, is presented on Fig. G1. The computing times are expressed in function of the number of data points, which are disposed on a regular time grid, in order to make a meaningful comparison. Confidence levels with the analytical approach are estimated with a 2-moment approximation. The number of samples for the MCMC approach is 10000 for the 95[th] percentiles and 100000 for the 99[th] percentiles. The smoothing coefficient is $\gamma = 0.5$, and the other parameters are default parameters of WAVEPAL. All the runs were performed on the same computer[12].

---

[12]CPU type: SandyBridge 2.3 GHz. RAM: 64GB.





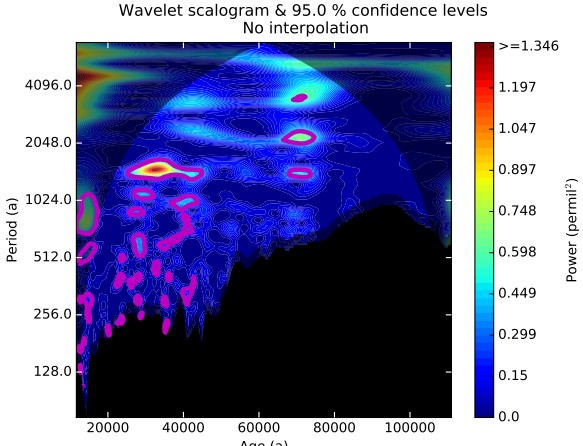

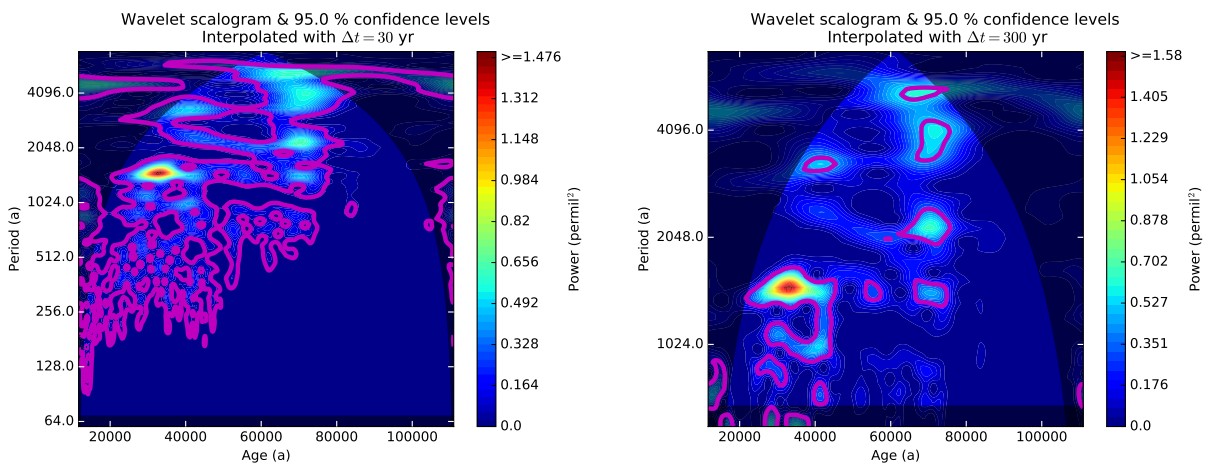

**Figure F2.** Scalogram of the time series presented in Fig. F1 and the 95 % analytical confidence levels against a red noise. (a) Raw time series. (b) Interpolated time series with $\Delta t = 30$ yr. (c) Interpolated time series with $\Delta t = 300$ yr.





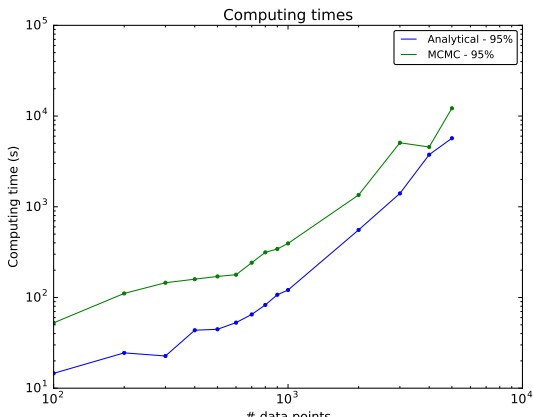 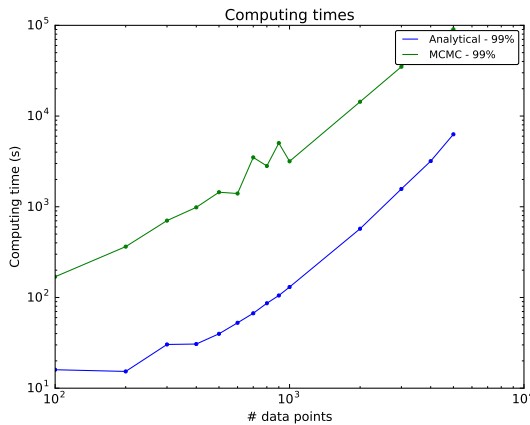

**Figure G1.** Computing times for generating the scalogram with analytical (blue) and MCMC (green) confidence levels, in function of the number of data points (disposed on a regular time grid). Log-log scale. Left: 95[th] percentiles. Right: 99[th] percentiles.

With this parametrization, and within this interval of the number of data points, we see that the analytical approach is faster than the MCMC approach. The analytical approach delivers computing times of the same order of magnitude whatever is the percentile (the two blue curves in Fig. G1a and G1b are in the same order of magnitude), unlike the MCMC approach, which must require more samples as the level of confidence increases, in order to keep a sufficient
5   accuracy. The difference between both computing times therefore increases as the level of confidence increases. Note, however, that the 2-moment approximation, for the estimation of the analytical confidence levels, is very fast from a computational point of view. Increasing the number of conserved moments may considerably increase the computational cost associated to the analytical approach. But this configuration is rarely used in practice because it often results in numerical instabilities and badly estimated percentiles, as explained in Sect. 4.2.

10   *Competing interests.* The authors declare that they have no conflict of interest.





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
