# Peer review of "A general theory on frequency and time-frequency analysis of irregularly sampled time series based on projection methods. II. Extension to time-frequency analysis"

_Nonlinear Processes in Geophysics, 2017_

## Author Comment (AC1) · 4 Jul 2017

Figures C1 and C2 are badly reproduced in the discussion paper. They are also available in the supplementary material:

- Figure C1 is in figures/heisenberg_boxes_Morlet.pdf

- Figure C2 is in figures/heisenberg_boxes_scale_discretization.pdf

---

## Referee Comment (RC1) · Anonymous Referee #1 · 10 Sep 2017

Review of "A general theory on frequency and time-frequency analysis of irregular sampled time series based on projection methods. II. Extension to time-frequency analysis" by Lenoir and Crucifix. This is a fine piece of work on wavelet analysis which will be useful for many geoscience areas which deal with irregular sampled data. I recommend to accept the manuscript for publication after my points have been addressed.

Recommendation: Minor revisions

[Figure]

This is the second part of a study on analysis methods for irregularly sampled time series.

1) Fig.1 shows a comparison between the method developed by the authors and a classical wavelet method for the NINO3 time series. I suggest to also compare the new method on the NINO3 time series which has been irregularly sampled by neglecting some values. That way the reader can be better see how well the new method does.

2) The author also include a trend component in their model. Trends are can be hard to identify. For example, which appears to be a trend in a time series could in fact be part of a very low-frequency oscillation. While no method probably can distinguish between these two cases it might be good if the authors would comment on this in the manuscript.

3) As in part I, most citations are in the form (author, year) even though they should be Author (year).

4) I suggest the authors discuss the form of the irregular sampling of the d18O data.

5) page 10, Line 25: Should "drived" be "derived"?

---

## Referee Comment (RC2) · Anonymous Referee #2 · 2 Oct 2017

This paper extended the idea proposed in Part I, and provided a general framework to do continuous wavelet transform for irregularly sampled time series. The idea is well presented. Algorithm is well explained with convincing numerical results.

I think the paper significantly contributes to the application of time-frequency analysis of irregularly sampled time series. I suggest it to be accepted for publication.

---

## Author Comment (AC2) · 4 Dec 2017

We are very grateful to the reviewer for the constructive comments and suggestions. We provide below a point-by-point reply to the reviewer's comments. In addition, during the review process, we got comments from other people that we judged pertinent to include in this revised version. The related minor changes are listed afterwards. When we mention a section or an equation, we refer to the new version of the

manuscript.

**1 Fig. 1 shows a comparison between the method developed by the authors and a classical wavelet method for the NINO3 time series. I suggest to also compare the new method on the NINO3 time series which has been irregularly sampled by neglecting some values. That way the reader can better see how well the new method does.**

We have added a scalogram of the time series for which $75\%$ of the data points were randomly removed. As the NINO3 time series does not hold prominent periodicities all along the time, we switched to another time series to better illustrate the method. We take the caloric summer insolation at $65°N$ over the last $2$ Myr, for which the scalogram exhibits clear periodicities in the precession and obliquity frequency bands. We then add some noise to this time series. See Sect. 3.2 for the modifications.

**2 The authors also include a trend component in their model. Trends can be hard to identify. For example, which appears to be a trend in a time series could in fact be part of a very low-frequency oscillation. While no method probably can distinguish between these two cases it might be good if the authors would comment on this in the manuscript.**

We now comment on this pertinent point in the two papers, in Sect. 3.3 of paper I and Sect. 3.1 of paper II. We have added the following sentence:
*Considering or not the presence of a trend in the model for the data is left to the user, given that we can always interpret a polynomial trend of low order as a very low-frequency oscillation.*

[Figure]

**3   As in part I, most citations are in the form (author, year) even though they should be Author (year).**

Corrected.

**4   I suggest the authors discuss the form of the irregular sampling of the d18O data.**

We have added the following sentence in Sect. 7.2 to draw the attention on the link between the sampling scheme and the form of the SNEZ:
*The form of the SNEZ, which is the black region at the bottom in Fig. 7 and 8, follows from the sampling of the time series presented in Fig. 6b.*

**5   page 10, Line 25: Should "drived" be "derived"?**

Changed to "driven".

**Other changes**

- Notations: All is bra-ket now, instead of a mix between bra-ket and bold symbols. For example, $\overline{\mathrm{sp}}\{\mathbf{a}\}$ is changed to $\overline{\mathrm{sp}}\{|a\rangle\}$.

- As in paper I, the angular frequency in the model for the data is now denoted $\Omega$. See Sect. 3.1.

- It turns out that there exists another published work than Foster's one about the wavelet scalogram for irregularly sampled time series, which basically suffers from the same limitations as Foster's algorithms. We mention this new reference in the introduction (Mathias et al., 2004), and discuss its formulas in appendix E.

- In Sect. 2, the continuous wavelet transform is now applied on functions which belongs to the Schwartz space, instead of the $\mathbf{L}^2$ space. That way, their Fourier transforms and the convolution product between two such functions are well-defined. Indeed, strictly speaking, the Fourier transform and the convolution product cannot be defined on $\mathbf{L}^2$. This is explained in appendix A, which is modified accordingly.

- The admissibility criteria in Sect. 2.3 is modified. Indeed, the zero-mean criteria that we wrote in the old version of the manuscript is a necessary but not a sufficient condition for the admissibility. The new condition is the general definition of the admissibility: $\int_{-\infty}^{+\infty} \mathrm{d}\omega |\widehat{\psi}(\omega)|^2 |\omega|^{-1} < \infty$.

- In Sect. 2.5, The so-called *Heisenberg uncertainty principle* is now called the *Fourier uncertainty principle*, since the former expression is an abuse of language in a non-quantum context. Other Heisenberg-like expressions are changed accordingly in appendix C.

- In Sect. 2.7, we have added a citation from Lilly and Olhede (2010) to motivate for the use of the wavelet ridges against the Hilbert transform. Here is the new paragraph:
  By construction, ridge filtering is well-adapted for filtering a multi-periodic signal, even if it is plunged in a noisy environment (Lilly and Olhede, 2010). In such conditions, it outperforms the techniques based on the Hilbert transform. As mentioned in Lilly (2010, p. 4135): "[...] the Hilbert transform can lead to disastrous results as the amplitude and phase will then reflect the aggregate properties of the multi-component signal."

- Correction of minor errors in Eq. (5), (38), (C3) and (C4).

- In Sect. 3.10, third item, $a_{\mathrm{max}}$ is changed to $a_{\mathrm{SNEZ}}$ since it turns out that both quantities are equal.

**References**

Lilly, J. and Olhede, S.: On the Analytic Wavelet Transform, IEEE transactions on information theory, 56, 4135-4156, doi: 10.1109/TIT.2010.2050935, 2010.

Mathias, A., Grond, F., Guardans, R., Seese, D., Canela, M., and Diebner, H.: Algorithms for Spectral Analysis of Irregularly Sampled Time Series, Journal of Statistical Software, 11, 1-27, doi: 10.18637/jss.v011.i02, https://www.jstatsoft.org/index.php/jss/article/view/v011i02, 2004.

---

## Author Comment (AC3) · 4 Dec 2017

We are very grateful to the reviewer for the comments. During the review process, we got comments from other people that we judged pertinent to include in this revised version. The related minor changes are listed in the reply AC2.

[Figure]

2017-27, 2017.

---

## Author Response (AR1)

**A general theory on frequency and time-frequency analysis of irregularly sampled time series based on projection methods. II. Extension to time-frequency analysis**

Guillaume Lenoir[1] and Michel Crucifix[1,2]

[1]Georges Lemaître Centre for Earth and Climate Research, Earth and Life Institute, Université catholique de Louvain, BE-1348, Louvain-la-Neuve, Belgium
[2]Belgian National Fund of Scientific Research, rue d'Egmont, 5, BE-1000 Brussels, Belgium

*Correspondence to:* Guillaume Lenoir (guillaume.lenoir@hotmail.com)

**This is the manuscript with the tracked modifications. See the interactive discussion for the replies to the reviewers.**

[revised manuscript text omitted]